# DSIF factor Spt5 coordinates transcription, maturation and exoribonucleolysis of RNA polymerase II transcripts

Krzysztof Kuś [1] ✉, Loic Carrique [2], Tea Kecman[1], Marjorie Fournier[1], Sarah Sayed Hassanein[1,3], Ebru Aydin [4], Cornelia Kilchert [4], Jonathan M. Grimes [2] & Lidia Vasiljeva [1] ✉

Precursor messenger RNA (pre-mRNA) is processed into its functional form during RNA polymerase II (Pol II) transcription. Although functional coupling between transcription and pre-mRNA processing is established, the underlying mechanisms are not fully understood. We show that the key transcription termination factor, RNA exonuclease Xrn2 engages with Pol II forming a stable complex. Xrn2 activity is stimulated by Spt5 to ensure efficient degradation of nascent RNA leading to Pol II dislodgement from DNA. Our results support a model where Xrn2 first forms a stable complex with the elongating Pol II to achieve its full activity in degrading nascent RNA revising the current 'torpedo' model of termination, which posits that RNA degradation precedes Xrn2 engagement with Pol II. Spt5 is also a key factor that attenuates the expression of non-coding transcripts, coordinates pre-mRNA splicing and 3'-end processing. Our findings indicate that engagement with the transcribing Pol II is an essential regulatory step modulating the activity of RNA enzymes such as Xrn2, thus advancing our understanding of how RNA maturation is controlled during transcription.

DNA-dependent RNA polymerase II (Pol II) is responsible for the transcription of protein-coding (mRNA) and non-coding RNA (ncRNA) in eukaryotic cells. ncRNAs represent a diverse class of transcripts including stable house-keeping ncRNAs such as small nuclear (sn)RNAs and unstable ncRNAs encompassing upstream transcripts derived from bi-directional promoters[1]. Both classes of Pol II transcription units (TU) undergo processing to become functional molecules. Precursor messenger RNA (pre-mRNA) processing takes place throughout the Pol II transcription cycle which consists of initiation, elongation, and termination stages. The coupling of pre-mRNA processing to transcription controls the timely recruitment of RNA processing factors as well as fidelity, efficiency, and regulation of the RNA-processing reactions[2–6]. Defective RNA processing is linked to neurodegenerative disorders and cancer[4,7–9], however, the

mechanistic understanding of how Pol II transcription controls RNA processing is limited.

Shortly after transcription initiation, pre-mRNA undergoes 5'-m7G capping, whereas splicing occurs during elongation. At the 3'-end of genes, pre-mRNA is cleaved and polyadenylated by the Cleavage and PolyAdenylation machinery (CPA). Recognition of the polyadenylation signal (PAS) and pre-mRNA cleavage by CPA is linked to Pol II dislodgement and termination of transcription[10–13]. PAS consists of AAUAAA and GU elements recognised by CPA and Cleavage Factors A and B, CFA and B in yeast and CFI and CFII in mammals[12–15]. The endonuclease subunit of the CPA (mammalian CPSF73 and yeast Ysh1) cleaves RNA between these two elements generating a 5'-monophosphorylated RNA end that is crucial for initiating 5'–3' degradation of the Pol II-associated RNA by exoribonuclease Xrn2. It was proposed that Xrn2 'chases' transcribing

[1]Department of Biochemistry, University of Oxford, Oxford, United Kingdom. [2]Division of Structural Biology, Wellcome Trust Centre for Human Genetics, University of Oxford, Oxford, United Kingdom. [3]Zoology Department, Faculty of Science, Cairo University, Giza, Egypt. [4]Institut für Biochemie, Justus-Liebig-Universität Gießen, Gießen, Germany. ✉e-mail: krzysztof.kus@bioch.ox.ac.uk; lidia.vasiljeva@bioch.ox.ac.uk

Pol II while degrading RNA ultimately leading to Pol II dislodgement from DNA (referred to as the 'torpedo' model)[16–19]. Although Xrn2 is an essential, highly conserved eukaryotic factor, the key aspects of the process leading to the termination of transcription, including the mechanistic understanding of the steps involved and how they are integrated into the context of transcription remain obscure.

In contrast to pre-mRNA, many ncRNAs do not undergo splicing, rely on the Integrator complex for the 3′-end formation and lack poly(A) tail[20]. Following endonucleolytic cleavage by Integrator, the exonucleolytic RNA exosome complex trims the 3′-end of the stable snRNAs or fully degrades transient species of ncRNA[20,21].

Recruitment of RNA processing factors during transcription is coordinated by the phosphorylation of the repetitive C-terminal domain (CTD) of Rpb1, the largest catalytic subunit of Pol II[4,22]. The CTD of Pol II is not the only repetitive sequence in which phosphorylation plays a crucial role in transcription. Spt5, a key factor for Pol II processivity, also has a repetitive C-terminal region (CTR) that undergoes phosphorylation by Cdk9[23–26]. Spt5, together with Spt4, forms the DSIF complex (*D*RB *S*ensitivity-*I*nducing *F*actor)[27–30]. Spt5 binds to Pol II after initiation and stays associated with Pol II throughout the transcription cycle[23,29,31]. Spt5 is functionally and structurally conserved between eukaryotes and bacteria, reflecting its important role in transcription[32,33]. Eukaryotic Spt5 and bacterial NusG share *N*usG *N*-terminal (NGN) and one *K*yprides, *O*uzounis, and *W*oes (Kow) domains. Spt5 has evolved a negatively charged N-terminal region and several additional Kow domains (Kow1-5 in yeast and 1–7 in humans) (Fig. 1a)[26]. The multidomain structure of Spt5 allows it to make extensive contacts with Pol II, nascent RNA, upstream DNA, and non-template DNA strand[34,35]. Spt5 can either support the paused state of polymerase or promote productive elongation by stabilising interaction around the DNA clamp[35–38]. Unphosphorylated Spt5 facilitates Pol II pausing downstream of promoters, recruitment of *N*egative *El*ongation *F*actor (NELF) and formation of the RNA clamp via its Kow4-5[39–43]. In higher eukaryotes, phosphorylation of NELF and Spt5 by Cdk9 is the key regulatory checkpoint during transcription leading to the dissociation of NELF and release of Pol II into elongation where Spt5 acts as a positive elongation factor[23,42,44]. Currently, how the phosphorylation of Spt5 contributes to Pol II processivity is not fully understood, but it may serve as a binding platform for transcription factors, like the CTD of Pol II. Indeed, recruitment of the PAF1 (*P*olymerase-*A*ssociated *F*actor 1) elongation complex is facilitated by the interaction of the Rtf1 subunit with Spt5-P[45,46]. On the other hand, Spt5 CTR is de-phosphorylated at the end of the genes by the CPA-associated PP1 phosphatase, which promotes the termination of Pol II transcription[23,47–50]. De-phosphorylation of Spt5-P is associated with deceleration of Pol II speed downstream of PAS. These events, according to the 'torpedo' model of transcription termination, are proposed to help Xrn2 to reach Pol II[48]. This model also assumes that the exonucleolytic degradation of RNA by Xrn2 precedes its engagement with the core of transcribing Pol II.

Using fission yeast as a model system, we demonstrate the formation of a stable complex between Xrn2, Rai1 (DXO in humans), DSIF (Spt4/5) and Pol II. We reveal that Spt5 is crucial for Xrn2 activation and recruitment, contributing to efficient transcription termination. We identify a region within Xrn2 that anchors Xrn2 to Pol II, facilitating its recruitment during transcription. Our findings suggest that Xrn2 engages with Pol II before RNA degradation. Transient transcriptome profiling (TT-seq)[51,52] reveals widespread transcription elongation defects upon Spt5 depletion. However, a subset of TUs accumulate TT-seq signal in the promoter-proximal region in parallel to a decreased read density in the gene body suggesting that Pol II is either retained at the promoter or undergoes premature transcription termination in the absence of Spt5. These genes are enriched in T-tracks downstream of promoters indicating that DNA sequences may promote premature transcription termination. In addition, lack of Spt5 leads to failed splicing, 3′-end processing and delayed transcription termination

implying that Pol II transitioned into elongation without Spt5 is not competent in supporting RNA processing. Our results emphasise the conceptual parallel between eukaryotic and bacterial mechanisms involved in transcription termination. Bacterial NusG is required for the recruitment and activity of termination factor Rho (RNA helicase)[53]. We conclude that Spt5 plays an important role in coordinating Pol II transcription and RNA degradation to ensure timely transcription termination and accurate gene expression. In summary, we provide insights on the 'torpedo' model where Xrn2 functions as a part of Pol II in collaboration with Spt5.

## Results

### Xrn2 forms a pre-termination complex with DSIF-Pol II

Spt5 is phosphorylated by Cdk9 on threonine 1 within its CTR composed of the repeats of a nonapeptide motif of the consensus sequence T[1]PAWNSGSK in *Schizosaccharomyces pombe* (*S. pombe*)[54]. To gain insight into how phosphorylation mediates Spt5 function in transcription, we undertook a proteomic approach. We pulled down a tagged Spt5 from the WT strain or cells lacking one of the PP1 phosphatase variants - Dis2 (*dis2Δ*). In parallel, we also purified mutated Spt5 where all threonine residues in CTR were either replaced by alanine (T1A) or glutamate as a phosphomimic (T1E). Surprisingly, analyses of the Spt5 purifications by mass spectrometry indicated a high enrichment of the components of the transcription termination machinery: Xrn2 and its interacting partner Rai1 in addition to known interactors of Spt5 such as Spt4, components of the RNA 5′-capping machinery (RNA-triphosphatase, Pct1 and guanylyl-transferase, Pce1) and subunits of Pol II (Fig. 1a, b - panel 1 and Supplementary Data 1). Enrichment of Xrn2 in all Spt5 purifications suggests that this factor interacts with Spt5-associated complexes in a phosphorylation-independent manner (Fig. 1b, panel 1). In contrast, components of the capping machinery, Pct1 and Pce1, were not detected in the purification of Spt5 T1E mutant (Fig. 1b - panel 2) in agreement with their specificity towards unphosphorylated Spt5[24,55,56]. Unexpectedly, Rtf1 reported to bind Spt5-P[25,45,46,57] and some subunits of the PAF complex were not present in either WT or T1E Spt5 purifications suggesting that the comparative proteomics approach may not capture the entire Spt5 interactome (Fig. 1b and Supplementary Data 1).

Next, we employed a fully defined in vitro system to test whether the Xrn2/Rai1 heterodimer interacts directly with Spt5 and Pol II. To this end, we purified Pol II and Spt4/5 (DSIF) (Fig. 1c, SDS-PAGE lanes 3, 4). The functionality of DSIF in stimulating transcription elongation was validated using a promoter-independent transcription elongation assay where Pol II was assembled with a dsDNA/RNA transcription bubble (Supplementary Fig. 1a–c and Supplementary Data 2)[58,59]. Analysis of the products of the transcription reaction demonstrated that Spt4/5 stimulated Pol II processivity and reduced the number of pausing events (Supplementary Fig. 1c, compare lanes 2 and 5) in agreement with the previously published work[60,61].

To study the nucleolytic activity of Xrn2, we expressed WT and catalytic mutant (Xrn2[M], D237A), in which a conserved aspartic acid was replaced with alanine (Fig. 1d)[17,18,62–64]. To obtain a high yield of stable protein, both constructs were expressed without the C-terminal part (1-885 aa of Xrn2) as previously reported[63]. The ability of the WT and mutated enzyme to degrade RNA was evaluated using 3′ FAM-labelled RNA with 5′-monophosphate mimicking the product of pre-mRNA cleavage by CPA complex (Supplementary Fig. 1d, P-RNA-FAM, Supplementary Data 2). Incubation of WT Xrn2 but not Xrn2[M] with the RNA substrate resulted in the gradual disappearance of the full-length RNA and accumulation of short degradation products (Fig. 1e, compare lane 1 to lanes 3–5). Using Xrn2[M] to avoid RNA degradation, we reconstituted a complex of Xrn2, Rai1, DSIF, and Pol II with a DNA-RNA scaffold representing an in vitro transcription elongation bubble[35]. Size exclusion chromatography confirmed that Xrn2/Rai1 stably associates with elongating Pol II-DSIF, forming a potential Pre-Termination Complex (PTC) (Fig. 1c, SDS-PAGE lanes 7–14).

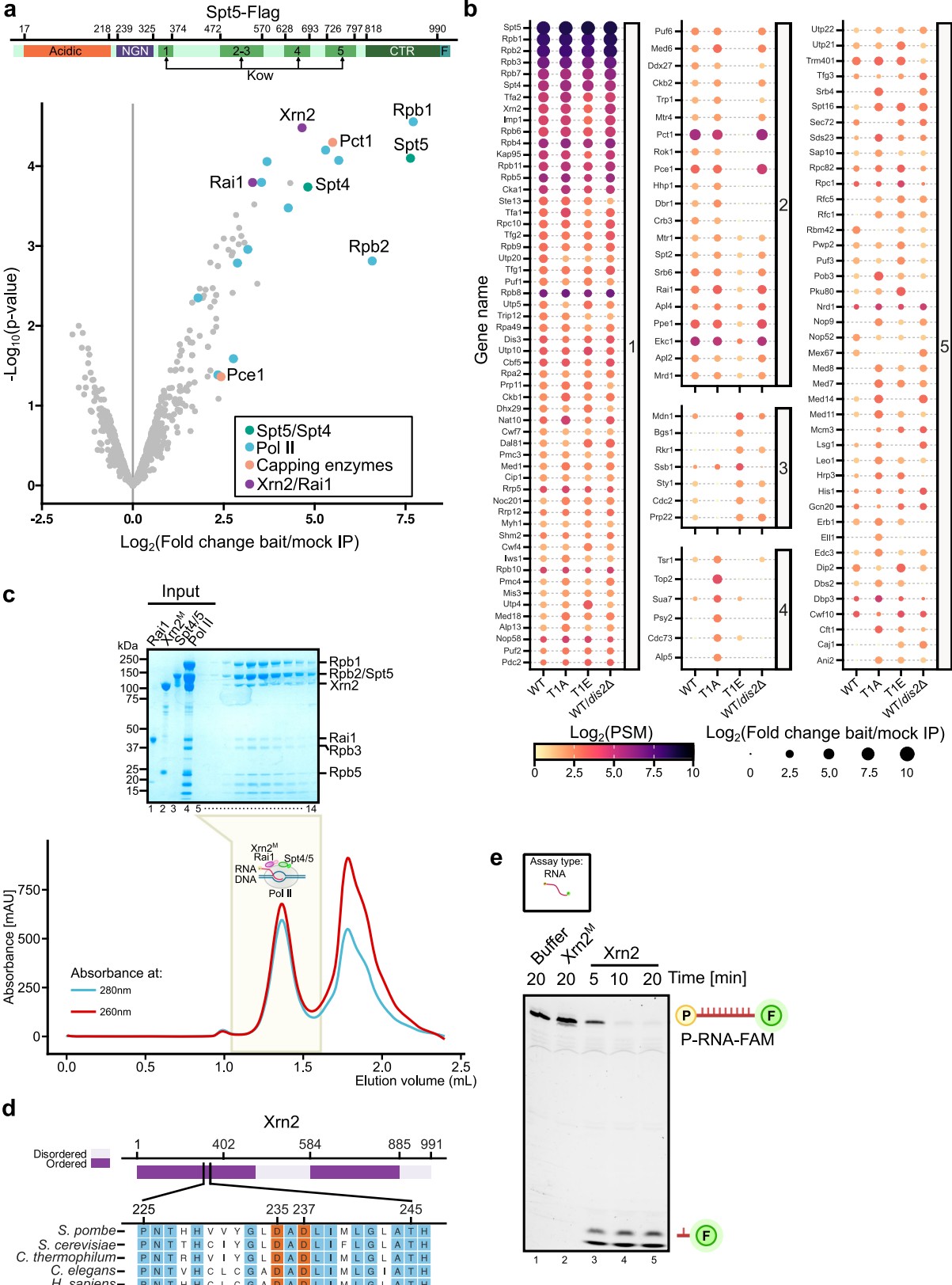

## Xrn2 directly interacts with the Spt5 and Rpb2-Rpc10 interface of Pol II

To further investigate if the interaction between Spt4/5 and termination factors depends on Pol II, we performed pull-down assays. Rai1 was immobilised and incubated either with Xrn2 alone or Xrn2+DSIF (Fig. 2a) revealing that Spt4/5 can bind termination factors independently of Pol II. To assess the DSIF requirement for the interaction of Xrn2 with Pol II, we prepared Pol II complexes on beads with or without Spt4/5 and nucleic acids (Supplementary Fig. 2a). This experiment revealed that Xrn2 can also interact with Pol II assembled on the DNA-RNA scaffold independently of Spt4/5 and Rai1 (Fig. 2b, lanes 9–10, 12–13). However, in the absence of a transcription bubble,

**Fig. 1 | Xrn2-Rai1 complex interacts with Spt5-Spt4 containing Pol II transcription complexes. a** Volcano plot highlighting proteins enriched in Spt5-Flag purifications. The top of the figure depicts the Spt5 domain organisation. Numbers correspond to the positions of amino acids. Statistical analysis was performed using empirical Bayes-moderated $t$-statistics ($n = 2$). **b** Xrn2 associates with Spt5 complexes in a phosphorylation-independent manner. Proteins enriched in purifications of Spt5 from WT Spt5, T1A, T1E and cells lacking PP1 (Dis2) phosphatase are presented as dot plots, and circle size reflects a number of peptides and colour corresponds to enrichment over mock IP. Proteins are grouped according to enrichment (panel 1 – present in all, 2 – decreased in T1E or *dis2*Δ, 3 – reduced in T1A or WT, 4 – more abundant in T1A, 5 – unassigned). Metabolic enzymes and ribosomal proteins were filtered out. **c** Xrn2-Rai1-Spt4/5-Pol II-DNA/RNA form a stable complex. Scaffold 1 used is listed in Supplementary Data 2 (compare Supplementary Fig. 2a) as previously published[35]. The reconstituted complex was subjected to size exclusion chromatography (Superose 6) and analysed by SDS-PAGE. mAU represents milli-absorbance units ($n = 2$). **d** Xrn2 alignment shows conserved catalytic residues that have been mutated in Xrn2^M (D237A for in vitro or D235A for in vivo experiments). **e** In vitro Xrn2 RNA degradation assay. 3′ FAM-labelled 5′-monophosphate-RNA (P-RNA-FAM, 26 nt) substrate was incubated with either Xrn2, Xrn2^M (D237A) or buffer for the indicated time. Intact RNA substrate and degradation products (indicated on the side) were resolved on 10% 8 M UREA-PAGE gel ($n = 2$).

the association of Xrn2 with Pol II required the presence of Rai1 (Fig. 2b, compare lanes 4 to 3 and 6 to 7). To further identify interacting interfaces within the PTC complex, we performed chemical crosslinking of the reconstituted Pol II-DSIF-Xrn2^M-Rai1-DNA-RNA complex with bis(sulfosuccinimidyl)suberate (BS3) (Supplementary Fig. 2b, lane 8) and analysed crosslinked peptides using mass spectrometry (Supplementary Data 3 and Supplementary Fig. 2c, d). A cluster of crosslinks is observed in the middle region of Xrn2 contacting Pol II (Rpb2, Rpb3 and Rpc10) as well as Kow5 and CTR of Spt5 (Fig. 2c). The region of Xrn2 engaged in interactions with Pol II is located between two parts of the enzyme that constitute its single catalytic domain. This region is not visible in the published high-resolution structure of Xrn2 with Rai1 and is predicted to be mostly unstructured[63]. To gain a deeper understanding of the interactions within the PTC complex, it was subjected to glutaraldehyde crosslinking (Supplementary Fig. 2b, lane 9) and cryo-electron microscopy (cryo-EM). We were able to obtain high-resolution 3D-reconstitutions of Pol II (at 2.67 Å, Supplementary Data 4 and Supplementary Fig. 3a) with nucleic acid scaffold and additional densities next to the RNA exit channel (corresponding to Kow5 of Spt5) and the region in the vicinity of Rpb2/Rpb10/Rpc10 (Supplementary Fig. 3b). In these reconstructions density corresponding to Rpb4/7 is missing (Supplementary Fig. 3b). Crosslinking data and AlphaFold[65,66] guided the modelling of the additional density, which corresponds to Xrn2 residues 481–496 (Fig. 2d and Supplementary Fig. 3b–d). We suggest that Xrn2 uses a small α-helical region embedded in the mostly unstructured segment to anchor the enzyme to Pol II. These interactions are mostly ionic with arginines (R492 and R496) of Xrn2 interacting with a negatively charged pocket formed by Rpb2/Rpc10 (Fig. 2d, e). In addition, isoleucine 489, located in the middle part of the Xrn2 peptide contacts a hydrophobic/negatively charged stretch of Rpb2 (Fig. 2e). Despite the low sequence conservation of the unstructured Xrn2 region, AlphaFold predicts that human and mouse Xrn2 enzymes contain small helical segments like the fission yeast counterpart with arginine residues present (Supplementary Fig. 3e) suggesting that this interaction may be conserved. Indeed, mutating positively charged residues or replacing the entire loop with a short glycine-rich linker in Xrn2 resulted in reduced interaction of Xrn2 with Pol II in vitro and compromised recruitment of Xrn2 during transcription in vivo despite higher expression levels observed for Xrn2^Δ444–555 compared to endogenous protein (Fig. 2f–h and Supplementary Fig. 3f, g).

Interestingly, the U1 complex of the splicing machinery binds to the overlapping region on Pol II (Fig. 2e)[67]. One of the ways U1-70K contacts Pol II is by insertion of positively charged residues (i.e., R121 and K118) into negatively charged pocket formed by Rpb2 protrusion and Rpc10 (Rpb12) but the orientation of the helix is rotated by 90° relative to the Pol II-interacting helix of Xrn2 (Fig. 2e). Therefore, splicing and termination machinery may compete for binding to the Pol II surface. In addition, modelling based on recently published structure indicates that the interaction of the Xrn2 catalytic core with Pol II is incompatible with Spt6 binding[68] (Supplementary Fig. 3h).

## Spt5 stimulates exonucleolytic activity of Xrn2

To further understand whether there is a functional link between Spt4/5 and Xrn2, we tested its effect on the 5′ to 3′ exoribonucleolytic activity of C-terminally truncated Xrn2 on 5′-monophosphorylated substrate with the 3′ FAM label (Supplementary Fig. 1d, P-RNA-FAM). Analyses of the reaction products revealed more efficient degradation of the substrate RNA by Xrn2 in the presence of Rai1 in agreement with the previous study[63] (Supplementary Fig. 4a, compare lanes 3–5 and lanes 12–14). Strikingly, the addition of Spt4/5 had a stimulatory effect on RNA degradation by Xrn2 (Supplementary Fig. 4a, compare lanes 3–5 and 15–20). Spt4/5 also stimulated Xrn2 activity on the RNA substrate associated with Pol II (Fig. 3a and Supplementary Fig. 4b, compare lanes 5 and 7 to 6 and 8). Taken together, Spt5 can stimulate Xrn2 exoribonucleolytic activity, suggesting that this effect can be relevant during transcription.

The observation that Xrn2 crosslinks to Spt5 in the vicinity of the Kow5 domain (Fig. 2c) prompted us to examine whether this region is sufficient for the stimulation of Xrn2 enzymatic activity. We purified Kow5-sCTR (residues 720 to 874) (Supplementary Fig. 4c) which was used in degradation assay based on fluorescence anisotropy (FA) (Supplementary Fig. 4d). Xrn2 degradation of the substrate P-RNA-FAM releases the fluorescent dye (FAM), and it can be monitored as a decrease of the anisotropy over time. The presence of Kow5-sCTR stimulated Xrn2 activity as the half-life of normalised anisotropy (a proxy for substrate half-life) decreased ~ 3-fold in the presence of the Spt5 construct (Fig. 3b). The stimulatory effect of Kow5-sCTR was comparable to the full-length Spt5 (Supplementary Fig. 4e). Since Spt5 interacts with DNA and RNA, we evaluated the binding of Kow5-sCTR to RNA using a fluorescence polarisation assay, which indicated that this construct has a modest affinity toward RNA (Fig. 3c). We conclude that Spt5 stimulates Xrn2 exoribonucleolytic activity via Kow5-sCTR.

## The exoribonucleolytic activity of Xrn2 is important for the dislodgement of Pol II from DNA and transcription termination

To examine whether Xrn2 activity directly contributes to the dislodgement of Pol II from DNA, we set up an in vitro transcription termination assay (Supplementary Fig. 4f)[58,69]. Here, Pol II was loaded onto a DNA-RNA scaffold and immobilised on streptavidin beads followed by incubation with Xrn2 WT or Xrn2^M. The Pol II dislodgement from the DNA template was monitored by assessing Rpb9 levels in the supernatant. Only WT Xrn2 led to the Pol II presence in the supernatant which correlated with RNA degradation (Fig. 4a, lanes 4, 5). In contrast, no Pol II release was observed upon incubation with Xrn2^M (Fig. 4a, lanes 6, 7) or RNase I_f which has endonucleolytic activity towards RNA (Fig. 4a, lanes 8, 9). Taken together, we confirm that the catalytic activity of Xrn2 directly contributes to Pol II dislodgement from the DNA template.

## Loss of the exoribonucleolytic activity of Xrn2 leads to profound global dysregulation of transcription

Next, we examined the contribution of Xrn2 catalytic activity to transcription. To generate a strain expressing Xrn2^M, we introduced an

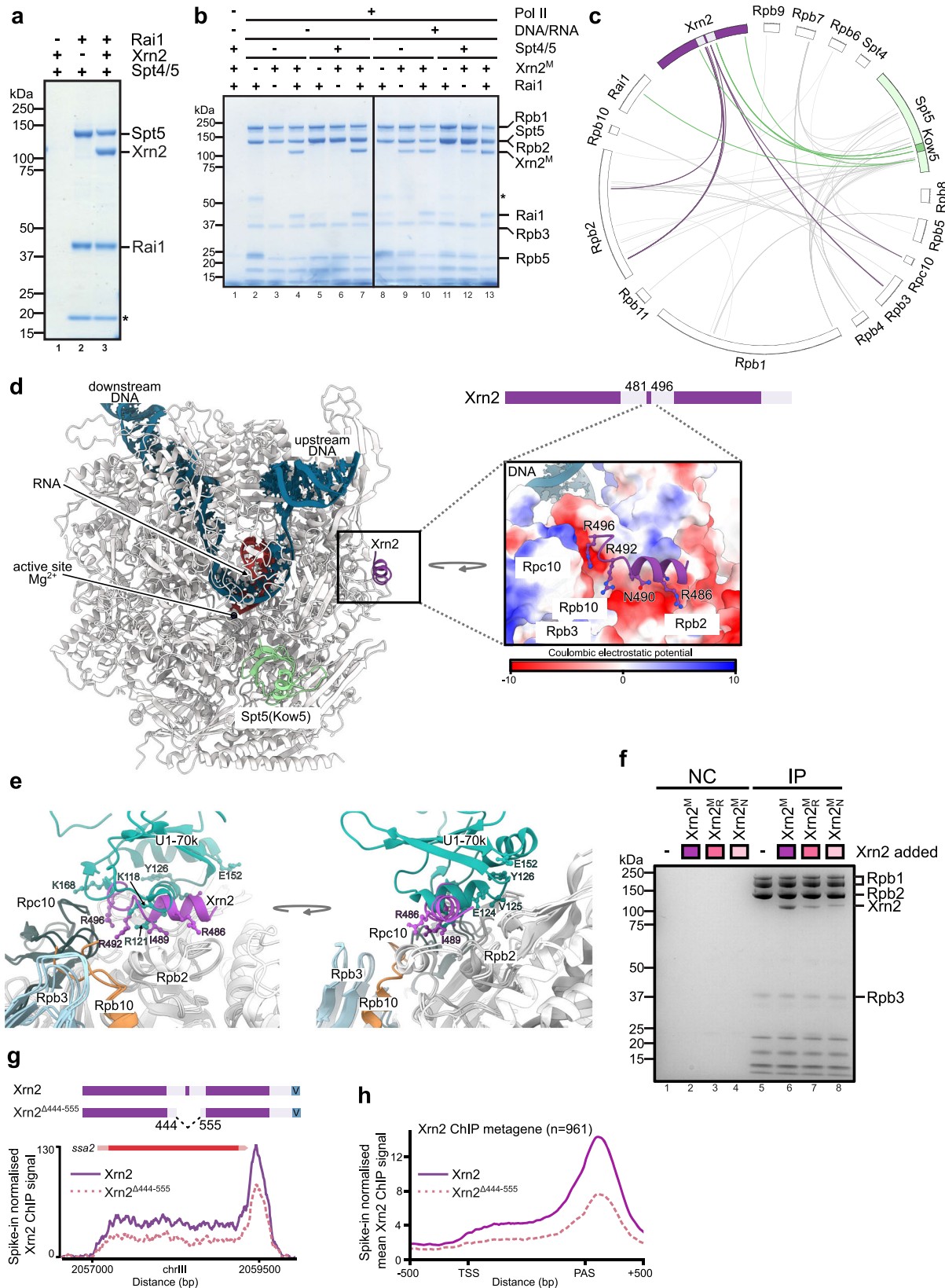

additional copy of the *xrn2* gene (V5 tagged) with the D235A mutation to the strain containing a degron form of WT Xrn2 protein (miniAIDx3-Flag-tagged)[70]. In parallel, we also generated a control strain that constitutively expressed WT V5-tagged Xrn2 to complement for auxin-depleted Xrn2 protein. We verified that V5-tagged WT and mutant Xrn2 proteins were expressed (Fig. 4b, lanes 3–6) and localised to the

cell nucleus (Supplementary Fig. 5a). AID-tagged Xrn2 protein was efficiently depleted after growing cells for 2 h in the presence of auxin (Fig. 4b). Xrn2 and its catalytic activity are essential for cell viability. Reintroducing WT Xrn2 but not Xrn2[M] can rescue growth after depletion of AID-tagged Xrn2 (Supplementary Fig. 5b). To facilitate the design of Xrn2 mutants in the future, we took advantage of this system

**Fig. 2 | Xrn2 interacts with Pol II and contacts Spt5 within the Kow5 region.**
**a** Spt5 interacts with Xrn2/Rai1. Rai1 was immobilised on beads and challenged with Xrn2 or Xrn2/Spt4/5. The asterisk indicates an unspecific degradation product ($n = 1$). **b** Xrn2 can interact with Pol II independently of Spt5. Pol II complex with or without nucleic acids was immobilised on beads via Rpb9-Flag (Supplementary Fig. 2a and Supplementary Data 2 - scaffold 1). Complexes were incubated with indicated proteins and beads bound proteins were resolved on SDS-PAGE. The asterisk marks an antibody-heavy chain. Lane 1 is a negative control for unspecific binding, where beads without immobilised Pol II were challenged with Xrn2$^M$/Spt4/5/Rai1 ($n = 1$). **c** Xrn2$^M$-Rai1-Spt4/5-Pol II-DNA-RNA scaffold complex crosslinking with BS3 coupled to mass spectrometry. Inter-protein crosslinks are shown as lines. Crosslinks between Xrn2 and Pol II are coloured purple and Spt5-Xrn2/Rai1 in green. **d** Cryo-EM model of the Xrn2-Rai1-Spt4/5-Pol II-DNA-RNA scaffold complex. RNA is shown in red, DNA in blue, the Spt5 Kow5 domain in green and the α-helical loop of

Xrn2, which docks enzyme to the negatively charged surface of Pol II (Rpb2/Rpc10) (zoomed area) in purple. **e** Xrn2 is anchored to the Pol II region which overlaps with U1-70 k binding surface from splicing machinery. Pol II-U1 snRNP complex (PDB: 7BOY)[67] was superimposed on the Pol II-Spt5-Xrn2 structure. Pol II subunits are depicted as follows: grey (Rpb2), orange (Rpb10), light blue (Rpb3), and dark grey (Rpc10). **f** Positively charged residues within the α-helical loop of Xrn2 contribute to Pol II binding. Pol II was immobilised on beads via Rpb9-Flag and challenged with buffer, Xrn2$^M$ (D237A), Xrn2$_R^M$ (D237A + R492E/R496E) or Xrn2$_N^M$ (D237A + all positive residues between 460–518 replaced by glutamic acid). NC - negative control (empty beads) ($n = 4$ for Xrn2$^M$ and Xrn2$_N^M$). **g** Region containing the α-helical loop of Xrn2 that binds Pol II contributes to exoribonuclease recruitment. Calibrated ChIP-seq results for Xrn2 or Xrn2 mutant (where region between 444–555 aa was replaced with a short glycine-rich linker) for representative locus. **h** Metagene of Xrn2 or its mutant recruitment (ChIP-seq) to coding genes ($n = 961$).

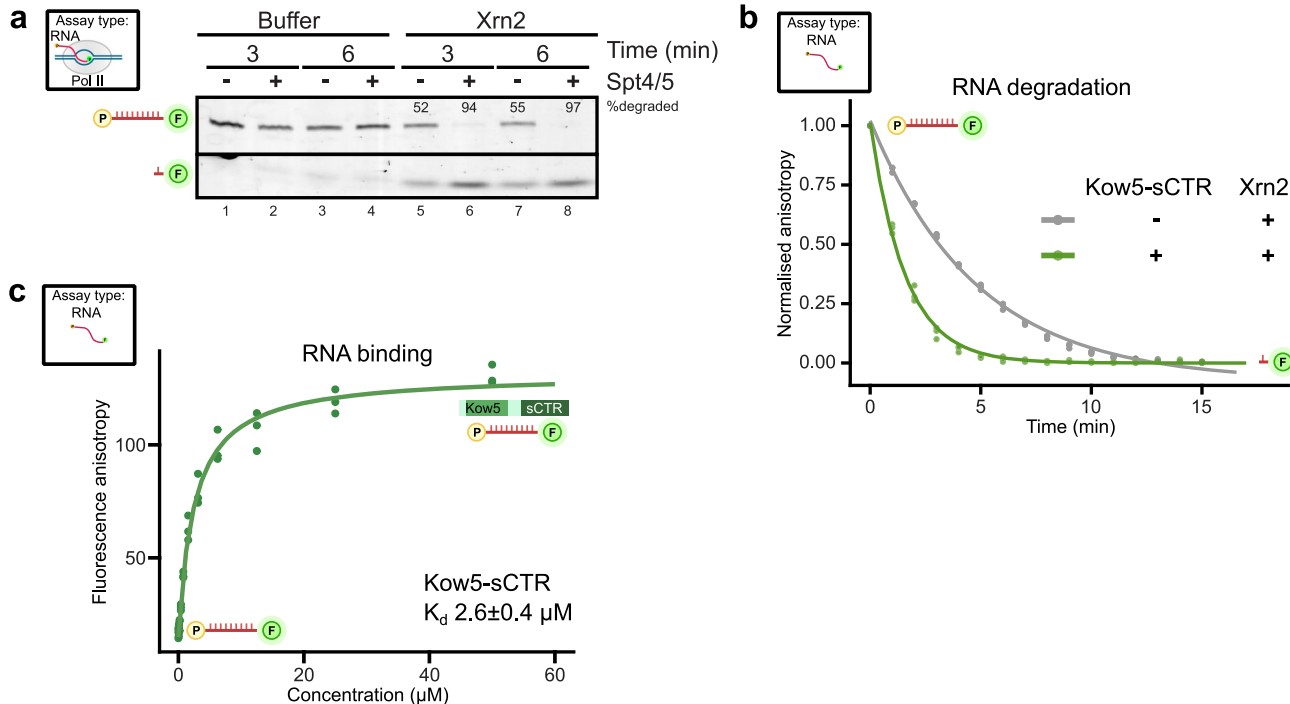

**Fig. 3 | Spt5 stimulates Xrn2-dependent RNA degradation. a** Spt5 can stimulate Xrn2 exoribonucleolytic activity towards 3′FAM-labelled 5′-monophosphate RNA (26 nt) in the context of the Pol II complex. Complexes were immobilised on beads using biotinylated non-template DNA (as depicted in Supplementary Fig. 4b). The extent of full-length RNA degradation is quantified for samples containing Xrn2 and indicated above RNA bands ($n = 3$). Source data are provided as a Source Data file. **b** Kow5-sCTR (Spt5 region 720 to 874 aa, N-terminal His-tag) stimulates degradation by Xrn2. Fluorescence polarisation anisotropy assay (as depicted in

Supplementary Fig. 4d) comparing RNA degradation kinetics of Xrn2 alone or Kow5-sCTR (compare to Supplementary Fig. 4e). Construct used with N-terminal His-tag. Source data are provided as a Source Data file. **c** The affinity of Kow5-sCTR to RNA was evaluated using a polarisation anisotropy assay. The 3′FAM-labelled 5′-monophosphate-RNA substrate was titrated with increasing amounts of purified protein. Construct used with N-terminal His-tag. Source data are provided as a Source Data file.

to map the location of the nuclear localisation signal (NLS) of Xrn2 by generating a series of deletions (Supplementary Fig. 5c). Removal of region between 444–575 aa, despite preserving a predicted NLS (396–418 aa)[71] interferes with protein localisation to the nucleus. Further analyses identified an additional low-scoring, putative NLS between 566–574 aa of Xrn2[72]. Indeed, a mutant lacking 444–555 aa shows nuclear localisation of Xrn2 (Supplementary Fig. 5c).

To initially assess Xrn2 catalytic activity in transcription termination, we used a colourimetric assay based on measuring the activity of the endogenous reporter phosphatase Pho1, which is regulated by premature termination of long ncRNA (lncRNA) *prt* (Supplementary Fig. 5d)[73–75]. Depletion of Xrn2 resulted in a 3-fold reduction in Pho1 activity, indicating compromised *prt* termination (Fig. 4c and Supplementary Data 5). This effect was reversed by WT Xrn2 but not Xrn2$^M$, highlighting the importance of Xrn2 catalytic activity. Having

established that Xrn2 RNA degradation aids Pol II dislodgement from DNA and efficient termination of *prt* transcription, we next investigated its global impact on transcription. We performed spike-in normalised transient transcriptome sequencing (TT-seq) in strains expressing either a single copy of AID-tagged Xrn2 or a second copy of the gene (WT or Xrn2$^M$) complementing auxin-depleted Xrn2-AID[51,52]. Successful incorporation of 4-thiouracil (4-tU) and RNA biotinylation were verified (Supplementary Fig. 5e). Analyses of the sequencing data revealed that Xrn2 depletion leads to an increase in the reads downstream of the annotated PAS, which is expected when either mRNA 3′-end processing, transcription termination or both processes are impaired. Profound changes in TT-seq signal were observed in Xrn2$^M$ similar to what is observed upon depletion of Xrn2 (Fig. 4d and Supplementary Fig. 5f). This includes the dramatic increase in the reads downstream of the PAS region demonstrating a global defect in

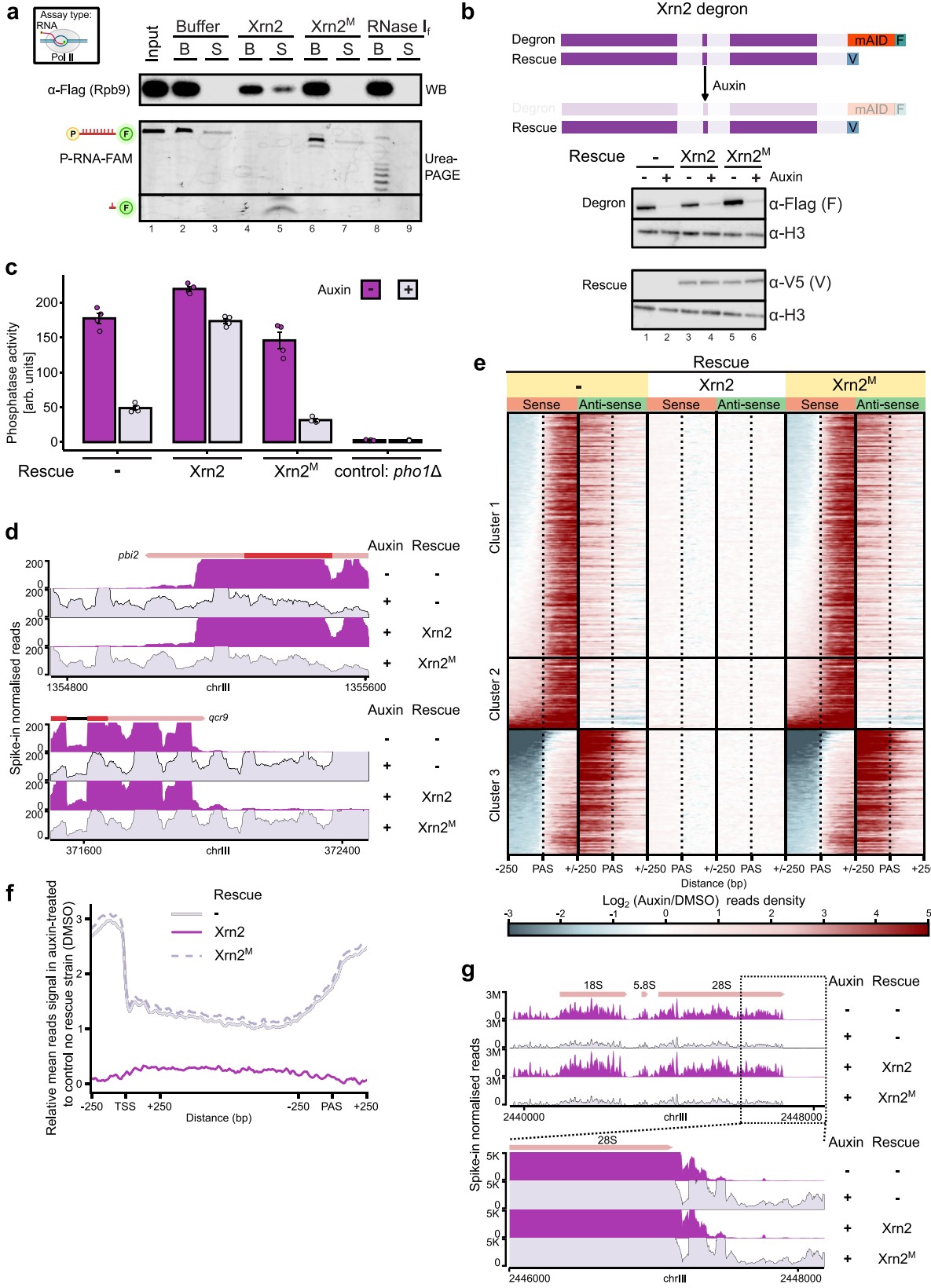

transcription termination in this mutant (Fig. 4d, e and Supplementary Fig. 5f, g). These observations are consistent with the published ChIP-seq data for Xrn2 D235A in *Saccharomyces cerevisiae* (*S. cerevisiae*) and analyses of nascent transcription based on bromouridine labelling, Bru-seq, when Xrn2 D235A was expressed alongside Xrn2 WT in human cells[17,62,64]. Global accumulation of the reads beyond PAS observed in

Xrn2[M] is accompanied by a decrease in TT-seq signal over the gene-body region for a subset of genes correlating with the increased transcription in the anti-sense orientation (Fig. 4e, clusters 1 and 3 and Supplementary Fig. 5h). In some instances, failed termination of transcription interferes with expression of downstream gene (Supplementary Fig. 5i). In contrast, there is class of genes with increased

**Fig. 4 | The catalytic activity of Xrn2 is indispensable for its function.**
**a** Degradation of nascent RNA by Xrn2 but not RNase I$_f$ dislodges Pol II. In vitro transcription termination assay was performed using Pol II immobilised on beads via biotinylated non-template DNA (schematic in Supplementary Fig. 4f). S - corresponds to the supernatant fraction, B – beads ($n = 3$, termination assessed by Western Blot). Source data are provided as a Source Data file. **b** Auxin-mediated depletion of Xrn2. Additional strains were created where WT or catalytically inactive Xrn2 mutant (Xrn2$^M$ - D235A) were introduced to complement depleted protein. Western blot analyses of AID-Flag tagged Xrn2 and constitutively expressing V5-tagged WT or Xrn2$^M$. Histone (H3) serves as a loading control. Source data are provided as a Source Data file. **c** Assessment of transcription termination using endogenous phosphatase activity-based reporter system (Supplementary Fig. 5d). Phosphatase activity of secreted acid phosphatase Pho1 serves as a proxy for termination efficiency. Xrn2$^M$ cannot complement the depletion of the WT enzyme. Data are presented as mean values +/− SEM ($n = 4$). Statistical analysis is included in Supplementary Data 5. Source data are provided as a Source Data file. **d** Functional

analyses of transcription upon acute depletion of Xrn2 or expression of catalytically inactive Xrn2 by transient transcriptome analysis (TT-seq). Representative snapshots demonstrating severe readthrough transcription in the absence of Xrn2 or expression of catalytically inactive Xrn2 (full signal Supplementary Fig. 5f). **e** Heatmap showing gene clusters differentially affected upon Xrn2 depletion or inactivation. Transcription on sense and anti-sense strands is presented as a log$_2$ fold ratio to control strain (DMSO treated strain without Xrn2 complementation). Cluster 1 and cluster 2 indicate that transcription downregulation correlates with readthrough on the opposite strand. Cluster 2 includes genes that appear as upregulated. Coding TUs that do not have another gene on the same strand within 250 bp upstream of TSS or downstream of PAS were selected for analysis ($n = 3190$)[146,149]. **f** Evaluation of genes in cluster 2 (Fig. 4e) presented as relative metagene. Accumulation of signal before TSS indicates that upregulation is caused by global readthrough. **g** Loss of Xrn2 activity results in transcription termination defects at Pol I transcribed rRNA genes.

---

signal in the 3' flanking regions and across the gene body (Fig. 4e, cluster 2) which can be attributed to readthrough as an elevated signal upstream of the TSS is present (Fig. 4f). Interestingly, loss of Xrn2 or its activity leads to an increase in readthrough transcription of ribosomal RNA (rRNA) suggesting that Xrn2 plays a role in the termination of Pol I transcription (Fig. 4g) which is consistent with previous work[76,77]. Overall, Xrn2 and its exoribonucleolytic activity are important for Pol II and Pol I transcription.

## Spt5 is required for restricting non-coding transcription and 'licencing' Pol II complexes at genic promoters

To study the direct contribution of Spt5 to transcription, we constructed a strain with a miniAID tag allowing rapid degradation of protein within 2 h (Fig. 5a)[70]. Spike-in normalised TT-seq experiments revealed that Spt5 depletion results in global reduction of RNA synthesis (Fig. 5b and Supplementary Fig. 6a, b) in agreement with the well-documented role of Spt5 as a transcription elongation factor[23,78,79] and severity of the defect correlates with gene length (Supplementary Fig. 6c). Interestingly, a subset of genes exhibits a non-uniform distribution of nascent RNA signal along the gene after Spt5 depletion. This class of genes is characterised by a high TSS/promoter-proximal signal relative to the low read density in the gene body ($n = 465$ coding TUs) (Fig. 5c, d) and might be subject to premature termination[62,80]. These genes show elevated T content with a tendency to form runs of poly-T (poly-U RNA) (Supplementary Fig. 6d). Gene ontology analysis revealed enrichment in categories related to stress and external stimulus responses (Supplementary Fig. 6e and Supplementary Data 6) which might indicate that genes in this category have a higher transcriptional plasticity.

Despite a prominent decrease in TT-seq signal for most genes, 1160 TUs show an increase upon depletion of Spt5. This group predominantly comprises TUs encoding for relatively short (median 770 bp compared to 1069 bp for full annotation), unstable transcripts such as ncRNAs (Supplementary Fig. 6b, f) generated from intra/intergenic regions, products of bi-directional promoters transcribed in anti-sense direction and housekeeping ncRNAs (small nuclear and small nucleolar RNAs) (Supplementary Fig. 6g, h).

In addition, Spt5 depletion shows splicing defects in agreement with the previous work suggesting that Pol II complexes escaping promoter-proximal checkpoint are not able to support efficient Pol II elongation and pre-mRNA processing (Supplementary Fig. 6i, j)[78,81–86]. We conclude that Spt5 participates in maturation and quality control ensuring functional coupling of transcription and RNA biogenesis.

## Spt5 is required for transcription termination

To explore the genome-wide role of Spt5 in transcription termination, we evaluated the TT-seq signal downstream of the annotated PAS analogously to Xrn2 depletion analysis. Striking accumulation of the

reads downstream of the analysed TUs was observed globally as well as for the individual TUs indicative of a readthrough transcription in Spt5-depleted cells compared to control (Fig. 5b, e, f and Supplementary Fig. 6k). Although the degree of the readthrough accumulation is milder in Spt5-depleted cells compared to what is observed in Xrn2 catalytic mutant (compare Fig. 5f to Supplementary Fig. 5g), this is consistent with Xrn2 activity being only partially reduced in the absence of Spt5 (Fig. 3a, b). The role of Spt5 in transcription termination is also supported by the results of the quantitative assay measuring the activity of Pho1 in the cells lacking Spt5. Deletion of *prt* reduces the effect of Spt5 depletion ruling out that the reduction in Pho1 activity is caused entirely by the elongation defect (Supplementary Fig. 6l).

Next, we assessed whether pre-mRNA 3'-end cleavage is affected in Spt5-depleted cells by RT-PCR using a primer pair that spans the PAS region for the representative genes *qcr9* and *spbc21b10.08c*. RT-PCR analyses demonstrated increased readthrough levels across the cleavage site upon Spt5 depletion compared to WT (Fig. 5g) suggesting that Spt5 is needed for 3'-end cleavage by the CPA. In addition, when we select PAS sites[87] and plot read density in a 10 bp window normalised to the gene body, we observe a significant shift in distribution (Fig. 5h). Interestingly, depletion of Xrn2 also exhibits reduced processing efficiency at PAS and splicing defects (Supplementary Fig. 6m, n). This is likely to be due to the extent of transcription interference resulting from the readthrough of upstream genes in Xrn2$^M$ (Supplementary Fig. 6o), which may prevent proper assembly of functional elongation complex and compromise all steps of pre-mRNA processing. In addition, we evaluated changes in Xrn2 and Pol II recruitment upon loss of Spt5 by ChIP-seq (Fig. 5i–k). Xrn2 association with chromatin is almost completely lost upon Spt5 depletion, and this effect remains pronounced considering correction for the changes in Pol II occupancy (Fig. 5k). Together, our data suggest that Spt5 is required for 3'-end processing of pre-mRNA as well as for the efficient degradation of the downstream cleavage product by Xrn2 and hence is a central player in coordinating transcription and maturation of pre-mRNA (Fig. 6).

## Discussion

In this study, we aimed to identify factors and mechanisms that underlie the function of the essential and conserved transcription factor Spt5. Notably, we demonstrate that Spt5 interacts with and enhances the exonucleolytic activity of Xrn2. Spt5 was shown to stimulate RNA cleavage by the Integrator complex[88], further supporting the emerging role of Spt5 in coordinating RNA processing and transcription. We propose that on functional Pol II that has passed through promoter-proximal quality control and is released into elongation, Spt5 functions to directly modulate the activity of enzymes involved in the RNA processing. We provide in vivo and in vitro evidence that Xrn2

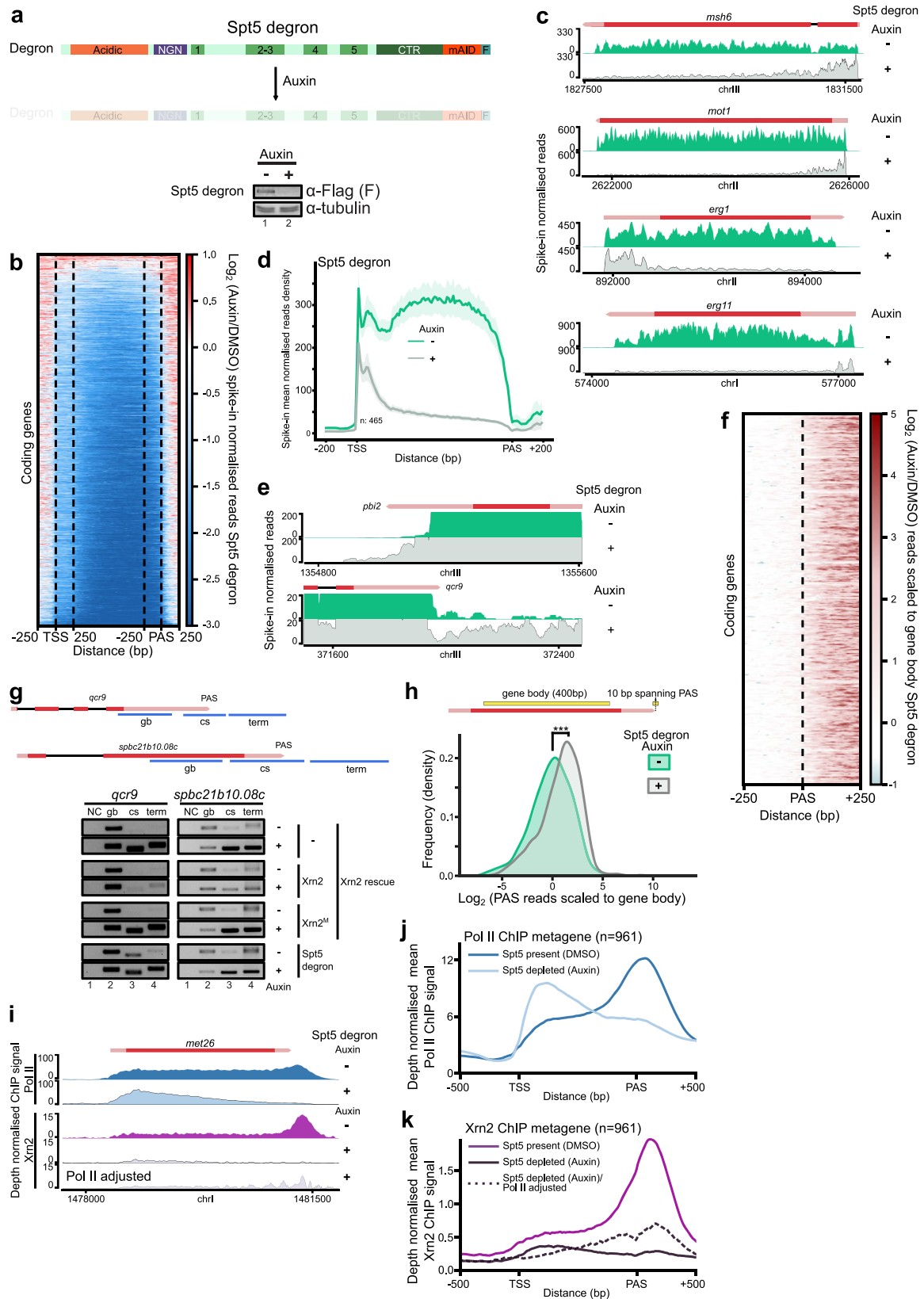

is tethered to Pol II through ionic interactions mediated by the residues from its flexible linker region and Rpb2/Rpc10 and upon formation of a complex with Pol II, it is subsequently activated by Kow5-CTR of Spt5. Our findings revise the model where degradation of nascent RNA by Xrn2 precedes its engagement with Pol II[17,18]. Although Xrn2 is present within the gene body region (in addition to the 3'-end of the gene), it

may not form a stable complex with Pol II during elongation since its binding to Rpb2/Rpc10 may be antagonised by binding of U1 snRNP[67]. Indeed, recently published studies of the reconstituted Xrn2(Rat1)-Pol II from yeast demonstrate that folded parts of Xrn2 bind Pol II near the RNA exit channel contacting Rpb1, Rpb2 and the stalk domain near the Kow5 domain of Spt5[68,89]. Our structure does not capture these

**Fig. 5 | Spt5 depletion affects multiple aspects of transcription including readthrough. a** Auxin-mediated depletion of Spt5. Western blot analyses of AID-Flag tagged Spt5. Tubulin serves as a loading control. Source data are provided as a Source Data file. **b** Heatmap of log₂ ratio in TT-signal between auxin and DMSO treated Spt5 degron strain. Gene body signal is scaled in the region between 250 bp after TSS and before PAS. **c** Example TT-seq genome tracks snapshot to illustrate 5′-end premature transcription termination/attenuation upon Spt5 depletion. **d** Metagene comparing profiles of preselected coding genes showing premature termination/attenuation before and after Spt5 loss (shaded areas indicate 95% confidence intervals). **e** Genome tracks highlighting transcription readthrough after Spt5 depletion. The full signal is shown in Supplementary Fig. 6k. **f** Genome-wide transcriptional readthrough on coding genes. The region around PAS is presented as a heatmap with a log₂ fold ratio of TT-seq signal for auxin, and DMSO treated Spt5 degron. Due to the global downregulation of transcription upon Spt5 loss, the read density was normalised to the gene body (for Xrn2 compared with

Supplementary Fig. 5g). **g** RT-PCR analysis of readthrough and cleavage in indicated strains. Gene body – gb, cleavage site region – cs and termination zone – term are depicted on the top panel. NC refers to the negative control (PCR for gb, without reverse transcription, *n* = 2). Source data are provided as a Source Data file. **h** Spt5 contribution to the 3′-end processing. Experimental PAS sites[87] were filtered to keep the most pronounced and the closest to gene end with PAS motif in 50 bp upstream sequence. Read density was calculated in a 10 bp window around PAS and normalised to the gene body density. The plot shows changes in distributions before and after Spt5 depletion. **i** Spt5 contributes to Xrn2 recruitment. Normalised Pol II or Xrn2 ChIP-seq signal is shown for representative locus in the presence or absence of Spt5. Additional track is included with Xrn2 signal adjusted for Pol II change between treated and untreated cells. **j** Metagene of Pol II distribution at coding genes (*n* = 961) before and after Spt5 depletion. **k** Metagene for Xrn2 distribution across coding genes (*n* = 961) before and after Spt5 depletion. In addition, the Xrn2 signal was adjusted for changes in Pol II occupancy after treatment.

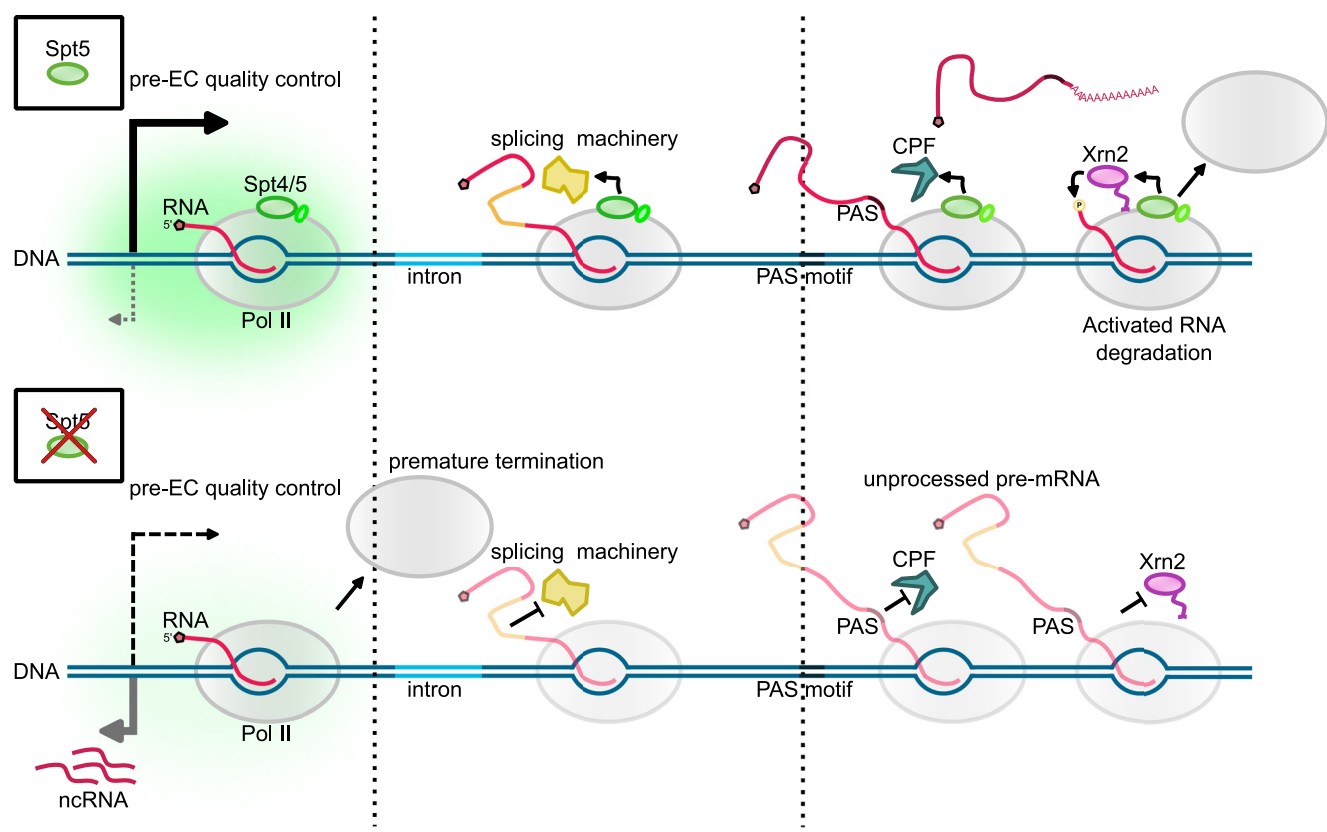

**Fig. 6 | Multifunctionality of Spt5 in transcription and RNA maturation.** Spt5 is required for quality control of pre-elongation Pol II complex (pre-EC) and its absence leads to premature termination affecting multiple genes. Spt5 plays a role in "licencing" Pol II for efficient co-transcriptional processing (including splicing and 3′-end cleavage). Xrn2 uses a short region to anchor itself to the core of Pol II.

Spt5 stabilises the Xrn2-Pol II complex and stimulates Xrn2 ribonucleolytic activity. Following endonucleolytic cleavage of RNA by CPA, Pol II bound Xrn2 degrades nascent RNA, as RNA gets shorter, Xrn2 pulls RNA out from Pol II leading to complex destabilisation and dissociation from DNA.

interactions, as we only observe a small α-helical region from the Xrn2 tethered to Pol II. Nevertheless, the published structure contains low-resolution unmodelled density in the similar region where we place the Xrn2 anchor. These structures led to a model where Xrn2 binding to Pol II antagonises the interaction of multiple Spt5 domains (except Kow5) as well as the binding of Spt6. One could envision that the engagement of Xrn2 with Pol II contributes to remodelling of the elongation complex facilitating transcription termination and 3′-end processing by slowing down Pol II. It is also possible that the recruitment of RNA processing factors to Pol II, such as Integrator[88], U1[67] and Xrn2 leads to structural rearrangements within Pol II where elongation factors are not fully dissociated from Pol II. On the other hand, depletion of either Spt6, Brd4, or PAF1 subunits leads to transcription

termination failure, suggesting that similar to Spt5, these factors may also contribute to the activity of RNA processing factors prior to being displaced from the Pol II complex[23,90–94]. Our cross-linking mass spectrometry data and biochemical experiments demonstrate that Spt5 contacts Xrn2 and stimulates its activity via its Kow5 and CTR domains. Although recent cryo-EM structures of Xrn2-Pol II report Kow5 being near Xrn2, there is no strong evidence of direct physical contact between these proteins[68]. However, CTR is not visible on the structures due to its flexible nature, and these structures may not capture all conformations of Xrn2-Pol II. Accordingly, the Spt5 mutant lacking Kow5 and CTR shows a global accumulation of the 3′-extended RNA in *S. cerevisiae*, suggesting that, indeed these regions of Spt5 are important at the 3′-end of the genes[95]. In line with this, depletion of Spt5 leads

to delayed termination in budding yeast and mammalian cells in agreement with Spt5 importance for timely termination of transcription[23,49,95,96]. Specifically, in the context of transcription termination, our data support a model where recognition of the PAS by the CPA triggers Spt5 dephosphorylation by PP1, which slows down Pol II. At the same time, Spt5 directly stimulates Xrn2 activity and contributes to its recruitment, assuring highly efficient transcription termination. Our discoveries provide the molecular basis for the previously reported accumulation of the 3′ readthrough upon depletion or small molecule inhibition of Spt5 in mammalian cells[23,97]. In addition, Xrn2 was enriched in Spt5 pull-down from mammalian cells, suggesting that the Spt5-Xrn2 link is conserved[23].

Furthermore, we show that loss of Spt5 leads to the accumulation of transcriptionally engaged Pol II at the promoter-proximal region, a severe reduction in transcription elongation/RNA synthesis rate across the gene body and an increase of 3′ readthrough observed for protein-coding genes. Accumulation of Pol II over the 5′ region of TUs was also reported by the previous study that employed NET-seq and ChIP-seq approaches to assess Pol II distribution upon depletion of Spt5[78]. DSIF can stabilise paused Pol II by facilitating the recruitment of NELF[98]. Phosphorylation of Spt5 and NELF by Cdk9 leads to the release of Pol II into elongation[23,44,49,99]. However, in organisms that lack NELF, Pol II pause release may not be strictly dependent on Cdk9[39]. Accordingly, depletion of NELF also alleviates dependency on Cdk9 in higher eukaryotes[100]. In addition, Spt5 can also control pausing independently of NELF[23,101,102], potentially representing a more ancient regulatory mechanism. Indeed, the NGN domain of Spt5 likely facilitates Pol II pausing through its interaction with the non-template strand of DNA, similar to the mechanism observed for NusG, the bacterial homologue of Spt5[32,103,104]. The choice between entry into productive elongation and premature termination is subject to regulation. Protein phosphatases such as PP1, PP2A and PP4 contribute to premature termination by antagonising Cdk9 activity on Spt5 and Pol II[105–108]. On the other hand, CPA-induced premature termination can be prevented by the U1 complex of the spliceosome by stimulating the Pol II elongation rate at A/T rich regions[109–112]. A/T-rich sequences can promote transcription termination by either mediating recruitment of CPA or by destabilising the Pol II complex and making them more susceptible to termination via backtracking-induced pausing[111,113–115]. Interestingly, we noted that the accumulation of Pol II at the promoter-proximal regions in Spt5-depleted cells correlates with the occurrence of T-rich stretches. This observation is consistent with the recent report demonstrating an increased termination of budding yeast Pol II on poly-U tracks[116]. In addition, transcription of poly-U stretches is known to terminate Pol III and this type of sequence bias might signify a strong dependence of Pol II on Spt5 for processive elongation[58,117–119].

In addition to the altered elongation rate, our TT-seq data also revealed that upon loss of Spt5, a fraction of Pol II that escapes into elongation shows defective pre-mRNA processing: splicing and 3′-end cleavage at the PAS. Reduced splicing efficiency when Spt5 is inactivated or depleted was also observed in other organisms[78,83–85]. These findings are consistent with the idea that Pol II undergoes a "maturation" process during the early stages of transcription and only a small fraction of Pol II is configured for efficient elongation and pre-mRNA processing[120]. At metazoan promoters, a large fraction of transcription initiation events is prematurely terminated[121–123] where two endonucleolytic cleavage complexes, CPA, and Integrator, cleave nascent RNA to mediate dislodgement of Pol II[20,107,124–126]. Upon depletion of the Integrator subunits, Pol II shows low processivity[127,128] and there is a widespread splicing defect[20], resembling the profile that we observe upon Spt5 depletion. Our data are consistent with the model where Spt5 contributes to quality control of Pol II complexes during the transition from initiation to elongation stages of transcription. Interestingly, an increased rate of transcription and an altered splicing landscape is associated with ageing further emphasising the importance of understanding how transcription is coordinated with RNA processing[129–131].

While depletion of Spt5 leads to decreased transcription elongation across most of the protein-coding genes, transcription of noncoding (nc) transcripts is increased. Previous studies demonstrated that depletion of the Integrator subunits leads to increased nc transcription suggesting that premature transcription termination plays a role in restricting ncRNAs[20,127,128]. Although simple eukaryotes such as yeast lack Integrator, mechanisms are in place that function to restrict nc transcription. In budding yeast, the Nrd1-Nab3-Sen1 complex restricts unwanted nc transcription, but this complex is not conserved[132–136]. Since nc transcripts do not usually undergo efficient splicing, Pol II complexes at nc promoters may resemble "misconfigured" Pol II and may not be licenced to enter elongation by Spt5.

The need for regulation of the enzymatic activity may be particularly important in the context of the functional coupling between transcription and post-transcriptional events. For example, in bacteria, where transcription and protein synthesis are spatially and temporally coupled, NusG interacts with and activates RNA helicase of termination factor Rho[53,137,138]. Although in eukaryotes, protein translation is spatially separated from transcription, it is coupled to RNA degradation, where cytoplasmic paralogue of Xrn2, Xrn1 plays a key role in co-translational degradation of mRNAs[139–141]. Despite the core of both exoribonucleases being highly conserved, the middle region of these proteins has undergone evolutionary diversification. Xrn1 uses this region to mediate interaction with ribosomes[141], whereas Xrn2 utilises it to provide a nuclear localisation signal and to anchor enzyme to Pol II (and potentially Spt5). The exact mechanism of how Spt5 and other cofactors regulate the activity of the Xrn-like enzymes remains to be understood in the future. We envision that Spt5 could facilitate the engagement of Xrn2 with the RNA substrate or allosterically stimulate its activity.

## Methods

### Strain construction and growth

*S. pombe* strains were grown in YES medium at 30 °C to $OD_{600}$ of 0.4–0.7 before harvesting. Standard homology-based methodology for genomic integration was used for either epitope tagging or gene deletion. Integrations were validated by Western Blotting with antibodies indicated in legends, which included: α-Flag (A8592, Sigma-Aldrich), α-V5 (V226, Sigma-Aldrich or MCA1360, Bio-Rad), α-H3 (07-690, Sigma-Aldrich) and α-tubulin (ab6160, Abcam). Xrn2 additional copy (rescue) was targeted to *ura4* locus under native promoter. The strains used in this study are listed in Supplementary Data 7.

### TT-seq and Bioinformatic analyses

The TT-seq experiments were performed twice as previously described with modifications[51,52]. Strains were grown in YES media (with reduced uracil to 10 mg/L) to $OD_{600} = ~0.5$ and cultures were treated with DMSO (dimethyl sulfoxide) or 1 mM auxin for 2 h at 25 °C. After depletion, cells were labelled with 5 mM 4-tU for 6 min, harvested and frozen in liquid nitrogen. Before RNA extraction OD of *S. pombe* was adjusted and mixed with a fraction of 4-tU labelled *S. cerevisiae* (spike-in, 100:1 ratio). RNA was extracted using the hot-phenol method. RNA (100 μg) was treated with DNase and fragmented using NaOH, followed by attachment of MTSEA-biotin-XX linker. Biotinylated RNA was purified using μMACS Streptavidin Kit. The specificity of the labelled RNA enrichment was assessed by comparing the pull-down enrichment of RNA from non-4-tU labelled cells. The size distribution of purified RNA was evaluated by a Bioanalyzer. Sequencing libraries were prepared using 50 ng RNA according to the manufacturer's recommendation for the NEBNext Ultra II Directional RNA Library Prep Kit for Illumina (NEB) and sequenced on the Illumina NextSeq 500.

Reads were quality controlled and trimmed using fastp software[142], aligned to a concatenated genome (*S. pombe* and *S.*

*cerevisiae*) using STAR[143]. Reads were split by species-specific chromosome names and *S. cerevisiae* uniquely mapped reads were obtained using SAMtools[144] and used for spike-in normalisation. Differential gene expression analysis was carried out using DESeq2[145] using the 2022 version of PomBase annotation[146]. Gene ontology enrichment was performed using a web-based server AnGeLi[147]. Metagenes and heatmaps were prepared with deepTools[148] (presented as a log$_2$ ratio and scaled to the gene body when indicated) using combined fission yeast genome annotation keeping only coding genes that do not have on the same strand in 250 bp before TSS or after PAS another transcription unit ($n = 3190$)[146,149].

The intron retention index was calculated as the ratio of intronic reads to surrounding exonic reads using a custom Python script based on PomBase annotation[146]. Selection of genes with premature termination in Spt5 depleted cells was performed using an in-house Python script and genes longer than 600 bp were considered. Read density was calculated in 300 bp downstream of TSS and divided by read density in the remaining part of the gene requiring at least 1.5-fold change ($n = 465$). Sequence bias in genes with premature termination was evaluated with seqPattern package in R. Potential cleavage defects were estimated as changes in read density spanning 5 nt around experimentally generated PAS sites[87] normalised to 400 bp read density in gene body (100 bp from PAS site). PAS sites were filtered to select the most prominent one and closest to the annotated coding transcript end and contain PAS motif (any of AATAAA, AATGAA, AATAAT, TAATAA, AAATAA, AATAAT, ATAATA, AATAAT) in 50 bp upstream sequence. Data was plotted as scaled frequency.

### ChIP-seq and analysis of RNA Polymerase II and Xrn2 occupancy

Cells were grown at 25 °C and treated with either DMSO or 1 mM auxin for 2 h. Prior to crosslinking, *S. pombe* cultures were spiked with *S. cerevisiae* SCC-6xV5 at a 250:15 OD ratio. Crosslinking was performed with 1/10 volume of CXF buffer (3:7 37% formaldehyde:CX buffer) for 20 min at 25 °C. The reaction was quenched with 1/7.5 volume of QS buffer (3 M glycine, 20 mM Tris) for 5 min. Cells were harvested, washed with cold TBS buffer (20 mM Tris-HCl pH 7.5, 150 mM NaCl), transferred to 2 mL tubes and frozen. On the day of the experiment, cells were washed with L05 buffer (50 mM HEPES-KOH pH 7.5, 150 mM NaCl, 1 mM EDTA, 1% Triton X-100, 0.1% sodium deoxycholate, 0.5% SDS) supplemented with protease inhibitors (Complete EDTA-free, 1 mM PMSF). Cell disruption was achieved by bead beating ($2 \times 9$ cycles of 60 s on/60 s off). The lysate was centrifuged at $180000 \times g$ for 20 min, 4 °C, and the pellet was resuspended in L01 buffer (similar to L05 but with 0.1% SDS) and centrifuged again. Chromatin was resuspended in L01 buffer and sheared using a Bioruptor Pico ($2 \times 35$ cycles of 30 s on/30 s off). The sample was centrifuged at 4 °C, $25000 \times g$ for 20 min. The soluble chromatin fraction was collected and snap-frozen in aliquots. Dynabeads Protein G were washed with TE buffer (10 mM Tris-HCl pH 7.9, 1 mM EDTA) and coated with antibodies (15 µL of Pol II antibody [8WG16, GTX20817, GeneTex] or 20 µL of mouse α-V5 tag antibody [MCA1360, Bio-Rad] per 60 µL of beads) for 1 h at 4 °C. Unbound antibodies were washed out with TE buffer. Chromatin samples were adjusted to 350 mM NaCl and incubated with the antibody-coated beads for 1 h with gentle mixing. Beads were washed three times with L01 buffer adjusted to 350 mM NaCl, followed by one wash with TE buffer. Elution was performed with EB buffer (50 mM Tris-HCl pH 7.9, 10 mM EDTA, 1% SDS) at 65 °C for 20 min. ChIP samples were diluted 2-fold with TE buffer and incubated with Pronase (1.5 mg/mL final concentration) for 1 h at 40 °C, followed by overnight de-crosslinking at 65 °C. RNase A and T1 were added, and samples were incubated at 37 °C for 1 h. DNA was purified using a ChIP DNA Clean & Concentrator kit and eluted in 20 µL of nuclease-free water. Libraries for Xrn2 and Pol II ChIP and input samples were prepared using NEBNext Ultra II DNA Library Prep kit with Sample Purification Beads following the manufacturer's instructions. Libraries were quantified by

Qubit and fragment size-estimated using a Bioanalyzer. Sequencing was performed on a NextSeq 500 platform. ChIP experiments were performed in duplicate. Raw sequencing reads were processed using methods similar to those employed in TT-seq analysis. For input subtraction, bigwig files were first normalised to sequencing depth, then this normalisation was reversed, and spike-in factors were applied. Only positive values were retained in the final bigwigs. For DMSO-treated samples, a common input was used for background removal (Fig. 2g and h). Due to the impact of auxin on the Pol II profile in *S. cerevisiae* cells (used as spike-in), comparisons for auxin-treated samples were made using normalisation to sequencing depth only (Fig. 5i–k). In addition, the Xrn2 signal was scaled by the ratio of Pol II signal change between DMSO, and auxin-treated samples. For the final analysis, bigwigs from two experiments were averaged. Protein-coding genes longer than 500 nucleotides were considered, and genes with Xrn2 ChIP signal at least two-fold above the average were included. The final dataset consisted of 961 genes meeting these criteria.

### Purification of native Spt5 complexes

Flag-tagged Spt5 was purified from WT as well as mutant cells lacking the *dis2* gene or strains containing T1A or T1E mutations in all the repeats of the Spt5 CTR[47]. As a mock control strain without tag on Spt5 was used. Cells were grown to OD 1.5, harvested and cell pellets were frozen at − 80 °C. Cells were disrupted in a freezer mill (SPEX SamplePrep) in liquid nitrogen. For purification, 5 g of cell powder was resuspended in 25 mL of lysis buffer (buffer L) (25 mM Tris pH 7, 100 mM NaCl, 0.5 mM MgCl$_2$, 0.5 mM β-mercaptoethanol) supplemented with 1 mM PMSF (phenylmethylsulfonyl fluoride), cOmplete Protease Inhibitor Cocktail (Roche) and phosphatase inhibitors: 1 mM NaF, 1 mM sodium orthovanadate, 2 mM imidazole, 1 mM sodium pyrophosphate decahydrate, 0.5 mM glycerol-phosphate, 5 µM cantharidin, 5 nM calyculin A. The lysates were incubated for 30 min in the cold room and cleared at $40000 \times g$ for 20 min at 4 °C. Following centrifugation, the lysates were incubated with 250 µL of an equilibrated α-Flag M2 agarose slurry for 1 h in a cold room. Beads were washed 10 times with 1 mL of buffer L, and proteins were eluted twice with 150 µL of 0.2 M glycine pH 2.5. Samples were neutralised with 25 µL of 1 M Tris. The samples were denatured with 8 M urea, reduced with TCEP, alkylated with 2-chloroacetamide, and digested with LysC, and trypsin. The reactions were stopped with formic acid and subjected to analysis by mass spectrometry. Results were analysed using the R environment. For the volcano plot, a protein was considered if present in both replicates of IP with a minimum of two PSM (peptide-spectrum match). Data were median normalised, and missing values in the mock experiment were imputed based on minimum value. Statistics were performed with the DEP package. Data for the mutants were plotted without normalisation as similar numbers of peptides were recovered in each sample and proteins were required to have at least 3 PSM and 3-fold change over mock IP.

### Pol II purification

Large-scale *S. pombe* Pol II purification was performed essentially as in ref. 47 with the following modifications. Cells were disrupted in a freezer mill (SPEX SamplePrep) or French Press (two passes at 35 kpsi). The cell lysate was incubated with α-Flag M2 beads for 1.5 h. Proteins were eluted with 5 mL of 1 mg/mL Flag-peptide (Sigma), followed by the addition of 20 mL of QA buffer (50 mM Tris pH 7.7, 5 mM NaCl, 10% glycerol, 0.5 mM MgCl$_2$, 0.5 mM Mg(OAc)$_2$, 1 mM β-mercaptoethanol). Protein eluate was then applied to an ion exchange chromatography column ($2 \times 1$ mL or $1 \times 5$ mL HiTrap Q HP, GE Healthcare) equilibrated with QA buffer. The column was washed with several column volumes of 8% buffer QB (same as QA, except 2000 mM NaCl) until a stable baseline was achieved. Protein was eluted with a gradient of QB buffer (up to 40%). Pol II buffer was exchanged to CB buffer (25 mM HEPES pH 7.9, 100 mM NaCl, 1 mM MgCl$_2$ and 1 mM β-mercaptoethanol) and

concentrated with Vivaspin 50 kDa MWCO (GE Healthcare). Pol II was aliquoted and snap-frozen in liquid nitrogen until the day of the experiment.

## Recombinant protein expression and purification

C-terminally truncated Xrn2 (residues 1–885) His8-tagged[63] and full-length Rai1 Strep-tagged were expressed from pRSFDuet plasmids in Rosetta *E. coli* strain, grown at 37 °C following induction with 0.3 mM IPTG for 12–15 h at 20 °C. Mutant variants of Xrn2 include: D237A (Xrn2$^M$), D237A/R492E/R496E (Xrn2$^M_R$) and mutant with multiple charge substitutions D237A/K460E/K475E/R479E/R486E/R492E/R496E/R501E/K511E/K518E (Xrn2$^M_N$). Cells were collected by centrifugation at 4 °C, 5000 × *g* for 10 min. For Xrn2, frozen pellets were re-suspended in NA buffer (50 mM Tris-HCl pH 7.5, 500 mM NaCl, 5 mM imidazole, 1 mM β-mercaptoethanol) supplemented with protein inhibitor cocktail, followed by lysis in French Press and addition of PMSF to 1 mM. Lysates were cleared at 4 °C, 40000 × g for 20 min, filtered and loaded onto a nickel-nitrilotriacetic acid resin and incubated for 30 min. Proteins were eluted with 200 mM imidazole. Fractions containing protein were concentrated and separated on HiLoad 16/600 Superdex 200 (Cytiva) equilibrated in CB buffer. Full-length Rai1-Strep was purified on Strep-Trap HP (Cytiva) column in CB buffer with 500 mM NaCl and eluted with desthiobiotin, followed by gel filtration in CB buffer using Superdex 200 (10/300, Cytiva). Proteins were aliquoted, snap-frozen in liquid nitrogen and stored at −80 °C. Full-length Spt4/5 heterodimer (Spt5 containing N-terminal His8-tag and thioredoxin) was expressed from pRSFDuet plasmid and purified as described before[47] with the exception that ion exchange purification was replaced by Heparin Sepharose (HiTrap Heparin HP Columns, Cytiva) and GF buffer was same as for Xrn2. Kow5-sCTR was expressed either as His-thioredoxin fusion (like Spt4/5 and used for degradation assay using fluorescence anisotropy) or N-terminally His-tagged (used for binding assay and degradation assay). Purification of these constructs was essentially identical to full-length Spt4/5 with the exception that gel filtration was performed on HiLoad 16/600 Superdex 75 (Cytiva). Plasmids used in this study are listed in Supplementary Data 7.

## Cryo-EM complex preparation

Pol II was mixed with RNA/template DNA at RT for 15 min. After, non-template DNA was added and kept at RT (scaffold 1, Supplementary Data 2). Spt5 was supplemented (~ 3-fold molar excess) and put on ice for 10 min. The complex was mixed with M2 Flag resin (150 μL of washed slurry) and incubated in the cold room for 30 min. Beads were washed four times with sample buffer (SB: 25 mM HEPES pH 7.9, 100 mM NaCl, 0.5 mM MgCl₂, 0.05% Tween) and incubated on ice for 30 min with an excess of Xrn2$^M$/Rai1. The resin was washed 5 times with 150 μL SB buffer and complex eluted two times with 35 μL of 2.5 mg/mL 3xFlag peptide (10 min incubation each). Sample buffer was exchanged against SB buffer using a 50 kDa amicon device. The complex was crosslinked with 0.1% glutaraldehyde for 10 min on ice and quenched with 8 mM aspartate and 2 mM lysine. Finally, the complex buffer was exchanged for the SB buffer as before. Grids were prepared using a Vitrobot mark IV (Thermo Fisher Scientific) at 100% relative humidity. Quantifoil Holey Carbon R2/1 200 mesh gold grids were glow discharged before applying 3.5 μL of sample at around 0.7 mg/mL and blotted for 3.5 s, blot force − 25 before vitrification in liquid ethane.

## Cryo-EM image collection and processing

Cryo-EM data were collected at the Oxford Particle Imaging Centre (OPIC), on a 300 kV G3i Titan Krios microscope (Thermo Fisher Scientific) fitted with a K2 Summit (Gatan) direct electron detector and a GIF Quantum energy filter (Gatan). Automated data collection was set up in SerialEM v3.8 and movies were recorded in counting mode. Data were collected using a 3x3 multi shots pattern, with a total dose of ~ 42.5 e⁻/Å², split across 50 frames, a calibrated pixel size of 1.05 Å/px

and a 20 eV slit. Sample-specific data collection parameters are summarised in Supplementary Data 4. Data were processed using the cryoSPARC V-3.X[150], following standard workflow. Pre-processing was performed using patch motion correction and patch-CTF estimation with default settings. Corrected micrographs with poor statistics were manually curated. A first round of blob picking was followed by a round of 2D classification allowing the generation of initial templates that were used for template picking. After 2D classification, the high-resolution classes were selected, and five ab initio models were generated and further refined using heterogeneous refinement. Only the class containing the DNA duplex and RNA was selected, and the 984000 particles were refined using NU refinement. After refinement of the CTF and correction of second-order aberration per optics group, the particles were refined a second time with NU-refinement before being exported to RELION 3.1[151] for further 3D classification. Based on crosslinking mass spectrometry results, we designed a mask surrounding most of the Pol II residues involved in these crosslinks and performed a 3D classification without alignment, using 10 classes and $T = 4$. After visual inspection, one class containing 215000 particles had an extra density for both the Kow5 domain of Spt5 and a fragment of Xrn2. The particles belonging to that class were re-imported into cryoSPARC, and a final NU-local refinement was performed, leading to a 2.67 Å resolution map.

## Structure determination and model refinement

Each of the individual subunits was first modelled by fitting the individual domains predicted by AlphaFold2 implemented in ColabFold[66,152]. In WinCoot[153], the restraints module was used to generate restraints at 4.3 Å and allow flexible refinement to fit the main chain into density. Multiple cycles of manual adjustment in WinCoot followed by real refinement in PHENIX[154] were used to improve model geometry. The final model geometry and map-to-model comparison was validated using PHENIX MolProbity[155]. All map and model statistics are detailed in Supplementary Data 4. Structural analysis and figures were prepared using UCSF ChimeraX[156].

## RNA degradation activity assays/RNA binding

Pol II was incubated with an excess of annealed RNA/template-DNA at RT and mixed with non-template DNA (scaffold 1a with P-RNA-FAM, Supplementary Data 2). This complex was mixed with equilibrated streptavidin beads (Streptavidin Sepharose High Performance, Cytiva) and incubated at room temperature for 30 min. Beads were washed with TB buffer (20 mM Tris, 40 mM KCl, pH adjusted to 7.9 with HCl), W500 (20 mM Tris, 500 mM NaCl pH adjusted to 7.9 with HCl) and again 3 times with TB buffer. Immobilised Pol II complex (20 μL) was mixed with either 10 μL SB buffer (25 mM HEPES pH 7.9, 100 mM NaCl, 1 mM MgCl₂, 1 mM β-mercaptoethanol) or Spt4/5 (or additional factor) in SB buffer and incubated for 10 min at RT. Beads were washed two times with TB buffer. The complex was mixed with buffer or Xrn2 and samples were collected after 3 and 6 min of incubation. Samples were resolved on 10% 8 M UREA-PAGE to evaluate RNA degradation and visualised for FAM fluorescence.

RNA degradation assays were also performed on RNA substrate (without Pol II). The reaction was set up in RB buffer (10 mM HEPES pH 7.9, 100 mM NaCl, 1 mM MgCl₂, and 1 mM β-mercaptoethanol) using P-RNA-FAM (Supplementary Fig. 1d and Supplementary Data 2). Aliquots were taken at 5, 10, and 20 min and reactions were terminated by the addition of stop buffer (10 mM EDTA, 3.5 M urea, 50 μg/mL heparin, 0.01% bromophenol blue, 0.015% xylencyanol in formamide) and resolved by 10% 8 M UREA-PAGE. Alternatively, the kinetics of RNA degradation were studied using fluorescence anisotropy (FA) assay. To this end, degradation of P-RNA-FAM (~ 250 nM) with protein (2–3-fold excess over RNA) or RB buffer was started by the addition of equal amounts of Xrn2 (optimised for the given reaction). Two constructs for Kow5-sCTR (Supplementary Fig. 4c) were tested without or with

thioredoxin (Fig. 3b, c and Supplementary Fig. 4e, respectively). P-RNA-FAM excitation was induced with linearly polarised light at 485 nm, and emission was measured both parallel and perpendicular to the plane at 520 nm at 25 °C using a PHERAstar FS plate reader (BMG Labtech). Polarisation anisotropy data were rescaled to 0–1 and analysed with in-house R script and fitted with an exponential curve. Kow5-sCTR (without thioredoxin) binding to 50 nM P-RNA-FAM was evaluated using fluorescence anisotropy assay with increasing amounts of protein. The binding curve was fitted as described previously[157] using the nonlinear least squares procedure in R.

### Termination assay
Pol II was incubated with an excess of annealed RNA/template-DNA at RT and mixed with non-template DNA (scaffold 1a, Supplementary Data 2). This complex was mixed with equilibrated streptavidin beads (Streptavidin Sepharose High Performance, Cytiva) and incubated at room temperature for 20 min. Beads were washed with TB buffer (20 mM Tris, 40 mM KCl, pH adjusted to 7.9 with HCl), W500 (20 mM Tris, 500 mM NaCl pH adjusted to 7.9 with HCl) and again 3 times with TB buffer. Immobilised Pol II complex (20 µL) was mixed with either 10 µL reaction buffer (RB) (25 mM HEPES pH 7.9, 100 mM NaCl, 5 mM MgCl$_2$, 4 mM ATP) or proteins in RB. Reactions were incubated for 10 min at 25 °C and supplemented with an additional 300 mM NaCl. Beads were spun down, and the supernatant was filtered through the Costar-X column to remove any potential beads. The remaining beads were washed with TB buffer. The reaction was inhibited with stop buffer (10 mM EDTA, 3.5 M urea, 50 µg/mL heparin, 0.01% bromophenol blue, 0.015% xylencyanol in formamide). Samples were used for Western blotting or UREA-PAGE to evaluate RNA degradation.

### In vitro transcription
Pol II complexes were prepared in a similar manner as for termination assay using scaffold 2 with $^{32}$P-RNA (Supplementary Data 2). After washes with TB buffer, the complex was incubated with buffer or Spt4/5. Transcription was started with the addition of 50 µM rNTPs and samples were collected at times specified. Products were resolved on 10% UREA-PAGE gel and visualised on FLA-7000 phosphoimager (Fujifilm).

### Immunofluorescence
Immunofluorescence was performed as previously described with minor changes[158]. Briefly, cells were grown in YES media to a final 0.5 OD and 9 mL culture was fixed with 1 mL 37% formaldehyde for 25 min at room temperature. Cells were washed twice with PEM (100 mM PIPES, 1 mM EGTA, 1 mM MgSO$_4$; pH 6.9) and resuspended in PEMS (PEM with 1.2 M sorbitol). The cell wall was digested with 0.25 mg/mL of zymolyase (100 T) at 37 °C for 1 h. Cells were permeabilised with PEMS + 1% Triton X-100 for 10 min and washed three times with PEM. Cells were then resuspended in PEMBAL (1% Bovine serum albumin, 100 mM Lysine hydrochloride, 0.1% NaN$_3$) and incubated with primary antibody (monoclonal α-V5-Tag mouse antibody /Elabscience, E-AB-20010/ and monoclonal α-Flag M2 mouse antibody /Sigma Aldrich/) and secondary antibody (AF488-conjugated goat anti-mouse antibody /Elabscience, E-AB-1056/). Cells in PBS were mounted on poly-lysine-treated coverslips (18 mm x 18 mm, 170 ± 5 µm high precision) using a DAPI-containing mounting medium (ROTI®Mount FluorCare DAPI) and images acquired on the Deltavision Ultra High-Resolution Microscope.

### RT-PCR
DNase-treated RNA isolated from indicated strains was reverse transcribed with either Superscript III or IV using primers listed in Supplementary Data 2 (three or one reverse primers were used for cDNA synthesis for gene body (gb) or cleavage site (cs)/termination zone (term) evaluation, respectively). PCR products were amplified using

Phire II polymerase with the following cycles: for the *qcr9* gene: gb fragments were amplified with 30 cycles, cs and term fragments with 28 for Xrn2 strains or 32 for Spt5 degron, whereas for *spbc21b10.08c*: 30 cycles were used for gb and 29 cycles for cs and term fragments.

### Phosphatase reporter assay
Cells were grown in YES media and transferred to EMM-P (no-phosphate media, NaOAc 1.2 g/L, L-Glutamic acid monosodium salt hydrate 5 g/L, glucose 20 g/L, mineral stock, vitamin stock, salt stock as for EMMG[159], adenine, histidine, uracil, leucine, lysine 225 mg/L). OD of cell cultures was adjusted, and cells were treated with DMSO or 1 mM auxin for 5 h at 25 °C. Cells were washed and resuspended in water. Phosphatase activity was measured as the ability of cells to dephosphorylate para-nitrophenylphosphate at room temperature for 10 min (the reaction was stopped with saturated Na$_2$CO$_3$ and measured at 400 nm). Phosphatase activity was normalised to OD, and for Spt5 degron it was scaled to DMSO-treated controls.

### Reconstitution of Pol II complexes and crosslinking
Complex formation between Xrn2$^M$, Rai1, Spt5, Spt4 and Pol II with nucleic acid scaffold (Supplementary Data 2, scaffold 1) was analysed by the size-exclusion chromatography (Superose 6 3.2/300), and Coomassie-stained PAGE.

For crosslinking, Pol II was incubated with ~2 molar excess of RNA/template-DNA at RT and mixed with non-template DNA (~3 molar excess, three different scaffolds were tested 1, 1b, 2, Supplementary Data 2). Excess of Spt4/5 and Xrn2$^M$/Rai1 were added. Complexes that were purified as described for cryo-EM or gel filtration were subjected to BS3 crosslinking on ice for 1 h. The reaction was quenched with Tris pH 7.5. Buffer was exchanged using an Amicon concentrator. Alternatively, the complex was assembled and crosslinked and then purified by Superose 6 3.2/300. Samples were denatured with 8 M urea, reduced with TCEP, alkylated with 2-chloroacetamide and digested with LysC, and trypsin. The reaction was stopped with formic acid and subjected to analysis by mass spectrometry.

### Analysis of pull-down or crosslinked peptides by mass spectrometry
Samples were denatured with 4 M urea (final concentration) dissolved in 0.1 M ammonium bicarbonate buffer before cysteines reduction with 10 mM TCEP for 30 min at room temperature, followed by alkylation with 50 mM 2-chloroacetamide for 30 mins at room temperature in the dark. Samples were then pre-digested with LysC (1 mg/100 mg of the sample) for 2 h before overnight digestion with trypsin (1 mg/40 mg of the sample). Tryptic digestion was stopped with 5% final formic acid. Digested peptides were centrifuged for 30 min at 12000 × $g$, 4 °C to remove undigested material. The supernatant was loaded onto a handmade C18 stage tip, pre-activated with 100% acetonitrile, by centrifugation at 1500 × $g$ at room temperature. Peptides were washed twice in TFA 0.1%, eluted in 50% acetonitrile / 0.1% TFA and speed-vacuum dried. Peptides were resuspended into 5% acetonitrile / 5% formic acid before LC-MS/MS analysis. Peptides were separated by nano-liquid chromatography (Thermo Fisher Scientific Easy-nLC 1000 or Ultimate RSLC 3000) coupled in line with a Q Exactive mass spectrometer equipped with an Easy-Spray source (Thermo Fisher Scientific). Peptides were trapped onto a C18 PepMac100 precolumn (300 µm i.d.x5 mm, 100 Å, Thermo Fisher Scientific) using Solvent A (0.1% Formic acid, HPLC grade water). The peptides were further separated onto an Easy-Spray RSLC C18 column (75 µm i.d., 50 cm length, Thermo Fisher Scientific) using either a 120- or 180 min linear gradient (15% to 35% solvent B (0.1% formic acid in acetonitrile)) at a flow rate 200 nl/min. The raw data were acquired on the mass spectrometer in a data-dependent acquisition mode (DDA). Full-scan MS spectra were acquired in the Orbitrap (Scan range 350–1500 m/z, resolution 70000; AGC target, 3e6, maximum injection time, 50 ms).

The 20 most intense peaks were selected for higher-energy collision dissociation (HCD) fragmentation at 30% of normalised collision energy. HCD spectra were acquired in the Orbitrap at resolution 17500, AGC target 5e4, maximum injection time 120 ms with fixed mass at 180 m/z. Charge exclusion was selected for unassigned and 1 + ions or + 2 ions. The dynamic exclusion was set to either 40 s or 60 s for 120- and 180-min gradients, respectively. Tandem mass spectra were searched using pLink software version 2.3.9. against an Uniprot *S. pombe* protein sequence database including common contaminants. Peptide mass tolerance was set at 20 ppm on the precursor and fragment ions. Data was filtered at FDR below 5% at the PSM level. Tandem mass spectra of crosslinked peptides were extracted using pLabel. Tandem mass spectra for the pulldown experiments were searched using Proteome Discoverer 1.4 against a UniProt *S. pombe* (ATCC 24843) database including common contaminants. The mass spectrometry proteomics data have been deposited to the ProteomeXchange Consortium via the PRIDE[160] partner repository with the dataset identifier PXD058168.

### Reporting summary

Further information on research design is available in the Nature Portfolio Reporting Summary linked to this article.

## Data availability

The model and map for the structure presented in this study have been deposited in the PDB database with the identifier 8QSZ and the EMDB database with the identifier EMD-18643. Previously published structures used in this study are 7B0Y, 7XN7, 3FQD and 8JCH. TT-seq and ChIP-seq data have been deposited in NCBI's Gene Expression Omnibus and are accessible through GEO Series accession numbers GSE244546 and GSE273510, respectively. The mass spectrometry proteomics data were deposited to ProteomeXchange with the identifier PXD058168. Source data are provided in this paper.

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

## Acknowledgements

We thank Nikolay Zenkin and Soren Nielsen for their help and advice on setting up an in vitro transcription system; Tomo Sugiyama and Japanese National BioResource Project (NBRP) for providing *S. pombe* strains for AID-tagging; Abigail Southers, members of the Vasilieva lab and Aleksander Szczurek for helpful discussions, advice, and valuable comments on the manuscript. This work was supported by the Wellcome Trust Senior Fellowship and BBSRC grants to L.V. (WT106994/Z/15/Z and BB/Y00194X/1), the Wellcome Trust Investigator Award to J.M.G. (200835/Z/16/Z and 222510/Z/21/Z) and the Emmy Noether Programme of the Deutsche Forschungsgemeinschaft (DFG) to C.K. (KI 1657/2-1). S.S.H. is funded by a scholarship grant (MM1/23/ID:1563097) from the Egyptian Mission Department. Access to electron microscopes was provided by the OPIC Electron Microscopy Facility funded by Wellcome JIF (060208/Z/00/Z) and equipment (093305/Z/10/Z) grants. Access to computational resources was supported by the Wellcome Trust Core Award Grant Number 203141/Z/16/Z with additional support from the NIHR Oxford BRC. The views expressed are those of the author(s) and not necessarily those of the NHS, the NIHR, or the Department of Health.

## Author contributions

K.K. and L.V. conceived, designed the experiments, and wrote the manuscript. K.K. generated strains and plasmids for this study, performed protein purifications, in vitro RNA degradation and in vitro transcription elongation and termination experiments, sample cross-linking for mass spectrometry experiments, RNA extraction, TT-seq library preparation and the bioinformatics analysis, participated in modelling of cryo-EM structure and mass spectrometry data analysis. E.A. and C.K. performed microscopy experiments to analyse the cellular localisation of Xrn2 mutants and quantification of the results. L.C. and J.M.G. performed cryo-EM experiments, including grid preparation data acquisition and assisted with data analyses. T.K. generated Cdk9/cyclin T1 (Pch1) baculovirus construct and purified Cdk9/cyclin T1 (Pch1) complex from insect cells. M.F. performed mass spectrometry experiments and assisted with analysis. S.S.H. helped with RT-PCR analysis. All authors edited the manuscript.

## Competing interests

The authors declare no competing interests.
