## [Peer Review file · Nature Communications]

DSIF factor Spt5 coordinates transcription, maturation and exoribonucleolysis of RNA polymerase II transcripts

Corresponding Author: Professor Lidia Vasiljeva

Version 0:

Reviewer comments:

Reviewer #1

(Remarks to the Author)

The manuscript by Kus et al, reports a new interaction between Spt5, Xrn2 and RNA polymerase II (PolII) as well as a stimulatory role of Spt5 on Xrn2 activity. They then pursue their analyses through the characterization of the role of Spt5 and Xrn2 in transcription regulation. While the initial findings reported are new and of interest, the present manuscript generally lacks in novelty, reporting mostly previously known roles of Spt5 and Xrn2. Additionally, the writing is lengthy and at times confusing, making it difficult to properly assess whether the authors are trying to develop new models or are barely describing previously established ones. For these reasons, further detailed below, the manuscript does not seem suitable for publication in Nature Communication.

Major points:

- While the reported Spt5-Xrn2-PolII interaction and the stimulatory role of Spt5 on Xrn2 activity are both new, these findings are not further investigated while instead experiments reproducing previously known results are privileged.

Experiments to consider could be to:

- investigate the impact of Spt5 on Xrn2 activity in vivo, or a more direct comparison of the phenotypes resulting from individual depletion of these factors.

- mutate the small helical region (481-496) of Xrn2 reported to mediate the interaction with Spt5, either in the R492/496 or I489.

- The results reported in Figure 3D, and the whole Figures 4 and 5 merely reproduce previously reported and well characterized roles of Xrn2 and Spt5. Curiously, the authors seem to be aware of this referencing previous literature.

- The manuscript is written in a very lengthy manner, in particular the introduction and the discussion being overtly lengthy. Focus would be good. The writing style would also profit from careful proofreading. Examples of such text related issues are listed in a specific section at the end of this document.

- While the authors conclude appropriately from the individual experiments, their broader interpretations (e.g. in the discussion) are sometimes overconclusive or too speculative.

For instance, in the discussion (l.645-646), the authors state that "We provide evidence that Xrn2 forms a stable complex with Pol II which supports the idea of Xrn2 travelling together with elongation complex". While they indeed establish the existence of an interaction between Xrn2 and PolII, none of their data suggest Xrn2 would be traveling with PolII.

Minor Points:

- Regarding transcription units (TUs) displaying increased gene body signal upon Spt5 depletion, the authors mention an effect of signal arising from upstream loci which can indeed be visualized in Figure 4E. Did they additionally check for TU lengths? Since a lack of Spt5 induces defects in TSS proximal pausing, polymerases might be able to run through short TUs while they would fail to elongate fully on longer loci, as reported upon depletion of the Integrator complex (Stein et al., 2022).

- "Spt5 is required for transcription termination" (l518). The authors seem to adhere to the previously proposed model that Spt5 is required for Transcription Start Sites (TSSs) proximal quality control and that in its absence, at some TUs, polymerases are not able to properly perform splicing (Figure 5G) or terminate a Transcripts End Sites (TESs) (l.543-545). Can this be directly called a role in transcription termination, as it could then be applied to many quality control mechanisms? This title would be more relevant if a direct dependency on the stimulatory effect of Spt5 on Xrn2 activity was established in this specific context, which is unfortunately not the case.

- Regarding the sequencing data library generation, the authors specify using a "NEBNext Ultra II Directional RNA Library Prep Kit for Illumina (NEB)" according to recommendation. This kit seems to have two possible workflows: either "polyA mRNA" or "rRNA depletion". Providing additional information on usage would be good. Indeed, the authors provide ncRNA based conclusions suggesting they did not rely on the "polyA mRNA" workflow but on the "rRNA depletion" one (as also seen in Figure S4E). However, Figure 4F, does show rRNA data with a track scaling of 3 million (while other TUs have a scale of a few hundred). Some leftover rRNA might be present, even after its depletion, but this difference in scale seems confusing. It also poses a risk of rRNA taking up most sequencing reads.

- In many instances where the TTseq signal is shown, it is truncated (Figure 4C, 5E, S4G, S5F, S5H, S5L), this should not be the case. Should the authors want to display the signal with its current scale, they also need to show the whole non-truncated signal (as in Figure 4F for instance). Alternatively, they may use log transformed signals to be able to visualize low signal.

- Figure S1C: the results text suggests that the processivity of PolII is increased, corresponding with reduced pausing events. This is not clear to me. Have the results been repeated and quantified?

- Figure S3A: Why is the input RNA still present after 20 minutes of Xrn2 (et al) incubations compared with the data in Fig 1E? Are these assays not identical?

- Inconsistencies in data representation; Figure 3A and C are the same assay. Why is the lower part of the gel (i.e. the decay product) shown in A and not in C? These experiments would also benefit from quantification, with replicate data.

- Figure 4B: adding some statistical measurement would be good.

- The order of results can be a confusing at times. E.g. catalytic activity of Xrn2 (in vitro, Figure 3D) should be linked to the in vivo data (Figure 4). The mapping of Spt5 in-between disrupts that flow.

- Lines 442-443 – 3' reads. Should this be 3'-extended reads? The data is not 3' RNAseq. This sentence was confusing.

- Spt5 rapid depletion TTseq 'highly reproducible'. This is based on only two replicates.

- Line 613 – I'm not sure if it can be said that 'transcription' is increased. TT-seq is a semi-transcriptional technique, but it is not a direct measure of transcription.

Text specific issues:

- The authors tend to overuse the definite article (THE) e.g. 'the core Pol II', 'the nascent RNA',...

- Some sentences seem surprising, e.g. the abstract statement: "Although importance of Pol II transcription in pre-mRNA processing is well established". Which RNA species requiring processing is supposed to exist without transcription? The authors are most likely referring to the coupling between transcription and processing, but this choice of phrasing is rather curious.

- In many instances, casual language is used, e.g. l.192 'Xrn2 chases down still transcribing Pol II', l.175 '... that Pol II requires help from Spt5...', ...

- Among instances of lengthy text, the authors tend to make additional introduction within the manuscript e.g. in Results section 1 (page 7) or in the

Reviewer #2

(Remarks to the Author)

The authors present a study in yeast showing new insights into the role of Xrn2 and Spt5, and how the latter drives the function of the former. The work aligns well with previous studies and contributes new findings that will be of interest to the gene regulation community. The biochemical work is beautiful and clean which is always a pleasure to see, and the overall presentation of the data is of high quality.

I have the following questions and comments to clarify some important aspects of the study:

1. The authors show that Xrn2 would compete for Pol II association with the U1 splicing complex. This means that Xrn2 would have to be dislodged during splicing, which for many genes with multiple introns, would occur many times during a single round of elongation. This means that Xrn2 cannot stay bound to Pol II all the time. The authors also suggest that the binding of Xrn2 may be transient. So, does Xrn2 really latch onto Pol II early on, or is it coming on and off at regular intervals or only during termination after CPA cleavage? This needs clarification.
2. Unless I am mistaken, the binding of Xrn2 would also compete with that of Spt6, based on structural work. If this is true, then Xrn2 binding would interfere with Spt6 activity which would be detrimental for normal elongation. It would not make sense then for Xrn2 to be Pol II associated all across the gene bodies (also due to the competition with splicing machinery mentioned above). In any case, it would be helpful to display the binding of Xrn in the context of the complete Pol II elongation complex structures available from the literature, to get a clearer picture of where Xrn2 binds via a vis others elongation proteins.
3. In the termination assay in Fig. 3D, the correct control to use would be another unrelated exonuclease that doesn't bind Pol II. This would be an important control to show that Xrn2 alone does the job but not another protein with the same RNase activity.
4. Lines 164-166: the authors should cite additional papers showing that depletion of Spt5 leads to termination defects (PMID: 29514850, PMID: 28318822).
5. The finding that Spt5 can suppress some non-coding transcription in yeast is very interesting. Could the authors speculate on what makes these promoters different from the others where Spt5 loss leads to decreased gene body elongation. Do they have high levels of Pol pausing at their TSSs? A Pol II ChIP would address this.
6. Is the mechanism by which Xrn2 terminates transcription known, meaning, how exactly does it displace Pol II once it reaches the RNA exit channel?

Reviewer #3

(Remarks to the Author)

Here Kuš et al., present a combination of biochemical, structural, and genomics data to study the role of Spt5 in transcription termination in *S. pombe*. First, the authors present a proteomics analysis of Spt5 interactions and show that the termination exonuclease Xrn2/Rai1 interacts with Spt5 regardless of its phosphorylation state. Next, they used a catalytically inactive mutant of Xrn2 to form a stable complex with Pol II, Spt4-5, and Xrn2/Rai1 and subject it to crosslinking-mass spectrometry and cryo-EM analysis. They describe density in their cryo-EM maps corresponding to a central Xrn2 helix that contacts the RPB2 subunit of Pol II and crosslinking data suggests that another Xrn2 region lies in proximity to Spt5 KOW5. They show that Xrn2 exonucleolytic activity is stimulated by Spt4-5 and particularly by KOW5-sCTR. The authors, then made an auxin inducible degron system to degrade Xrn2 and complement either with WT or catalytically inactive Xrn2. They observe run-on transcription after PAS signal, as previously observed. The defect is efficiently restored by WT Xrn2 but not with the catalytically inactive version of the protein. Finally, they transiently degrade Spt5 and find several defects in Pol II transcription, including enhanced promoter-proximal accumulation and defects in termination, as well as defects in splicing. The authors conclude, based on their experiments, that Xrn2 forms a stable complex with terminating Pol II (called pre-termination complex, PTC) that enhances Xrn2 exonucleolytic activity and degrades RNA in a 5-3' direction causing Pol II dislodgement and recycling.

The authors present a collection of observations that are consistent with their model, however, their study lacks key controls and in some cases the observations are not clear, or the effects are marginally different from their controls. The manuscript could be divided into two different stories, and this would likely make the story more digestible. Specifically, the overlap between the structural and biochemical data and the genomics data is limited. If the authors could better bridge these two sets of experiments through the creation of specific mutants, this would support the fusion of the two sets of results. This manuscript could be appropriate for Nature Communications after major revisions.

Major comments

- 1- The introduction and discussion sections are quite long. The manuscript would be greatly helped if the authors trimmed these sections to focus on key points that are relevant to this manuscript. The introduction also orders the findings (final two paragraphs in the introduction) in the opposite order that they are presented in the manuscript. It would be clearer if both orders were consistent.
- 2- The Xrn2 degradation assays are displayed in several ways across the manuscript (e.g., gel-based assays, anisotropy), making it difficult to directly compare experiments. It is also difficult to observe stimulation in their current setup because RNA digestion intermediates are not visible in their gel based assays (either full RNA or the fully degraded RNA). Densitometry analysis of gels would help make differences between experiments clearer. Sometimes the degradation product is not shown (eg figure 3C), and the time points used between experiments are inconsistent. It is also unclear to the reviewer how many times these experiments were repeated. Consistency between the experiments would greatly enhance interpretability of the experiments.
- 3- The analysis of XL-mass spectrometry data is incomplete. Cutoffs for XL-mass spectrometry are missing in the supplemental figures, specifically mapping crosslinks onto a Pol II structure to see how many are satisfied by the distance of BS3. Xlink Analyzer is a good tool to use to visualize this. A few representative spectra for the described Xrn2-Pol II crosslinks would also be helpful to provide to show the quality of the data.

4- Cryo-EM data presentation is incomplete. Please include data processing tree, angular distribution plot, FSC curves, local resolution plot of maps used to build the structure, a representative micrograph with scale bar, and representative 2D classes with scale bar. It is likely that the authors did not fully classify their data to find Xrn2 containing particles. From the data analysis description, it appears that no specific classification approach was used to specifically classify for Rpb4/7 or Xrn2. Subtracted classification or focused classification on low pass filtered maps can often yield better results. Here the authors only classified based on their crosslinking data locations.

5- Figure 2, Panel E, U1 and Xrn2 are not binding Rpb2 same way. When superimposed, they lie perpendicular to each other. This needs to be reworded. It is possible to say that both proteins use overlapping binding surfaces. It looks like Xrn2 is binding the Rpb2 protrusion. If true (or another Pol II domain), this needs to be more specifically described using Pol II structural nomenclature.

6- The authors perform no experiments to confirm their cryo-EM results. Mutations in the Xrn2 helix that contacts Rpb2 should disrupt the interaction with Rpb2 and should result in reduced Xrn2 activity in the termination and/or RNA degradation assays. If such mutations are identified, it would be useful to add them into their genomic assays. This would also help bridge the genome wide and biochemical findings.

7- The authors claim that CDK9 does not affect Xrn2 activity in their assays. Have they tested whether CDK9 is indeed phosphorylating Xrn2? Mass spectrometry analysis would address this issue. Additionally, the authors could make a phosphomimetic mutant based on previous data from the Fisher lab (10.1101/gad.269589.115) as a control.

Minor comments

1- The authors refer to Spt4-5 as Spt5/4. This is a bit awkward and is non-standard in the field.

2- Figure 1- some interactors are missing in the IP experiment (e.g., Rtf1). The authors mentioned that is due to a transient interaction, however there is no indication of what the authors consider transient, as some factors are known to be stably bound to Pol II and are still missing in their experiment. The reviewer suggests omitting any reference to transience.

3- The authors claim that the degradation we see is a 5'-3', as expected for the previously described Xrn1 activity, but there is no control for that assumption. A protected 5' RNA would be a useful negative control for this point.

4- Figure 2, pulldown/IP experiments. It would be good to see the inputs at least in the supplemental figure. Also, some IP combinations are missing, for example Rai1 + Spt4/5. This will help to rule out any unspecific binding to the beads. Also, more labelling of the gels would be helpful (like labelling the position of Rpb1, Rpb3, Rpb5, Spt4, etc.)

5- The discussion does not compare their findings in light of alternative models. For example, how are these findings consistent or inconsistent with previous studies supporting the torpedo model?

Version 1:

Reviewer comments:

Reviewer #1

(Remarks to the Author)

We thank the authors for their rebuttal and revisions to their manuscript. For the most part, they have addressed all our concerns in regard to data with newly performed experiments supporting their conclusions appropriately.

However, one major point remains that the writing is very difficult to follow in places – especially in the Introduction and Discussion sections which are incredibly lengthy. The introduction should be much more concise and does not need to introduce things that are not included in the rest of the manuscript. For example, RPB1-CTD phosphorylation has 13 lines in the introduction but is not mentioned anywhere else in the manuscript. Additionally, lines 111-142 describe the findings of the results but includes far too much detail for an introduction summary. Overall, this still needs to be made much more concise.

Further, we made the point of the overuse of the definite article (THE) throughout the paper e.g. “The RNA processing factors”. This still needs attention because it has been removed in places where it should remain and vice versa. Below are a few examples, but it should be carefully addressed throughout the manuscript.

Examples where THE should be removed;

Line 43-44; “the bi-directional promoters”

Line 68; “The Integrator”

Line 70; “The RNA processing factors”

Line 96: “The promoters”

Examples where THE should be included;

Line 161; “In the purification of the Spt5 T1E mutant”

Line 166; “...to test wheter the Xrn2/Rai1 heterodimer”

Finally the use of “the RNA enzymes” in the abstract is a bit cryptic and should be reworded. Does this simply mean XRN2?

Reviewer #2

(Remarks to the Author)

The authors have addressed my questions reasonably. I have no major concerns.

Reviewer #3

(Remarks to the Author)

I appreciate the efforts the authors have taken to address my concerns. There are some outstanding issues that should be addressed prior to publication.

1- Figure 2D, E- Labelling on RNAPII is sparse. Labels are needed for upstream and downstream DNA, Spt5 (not just KOW5), the active site Mg²⁺ needs to be added to orient readers, and different colors/shadings would help in Panel E to distinguish between the indicated RNAPII subunits that are highlighted. Currently, labels are provided for 4 subunits, but all subunits are colored white, and it is hard to know which ones are which.

2- Figure 3C- show in the same manner as Figure 3A (currently, 3C is missing the nucleotide level band (bottom panel in 3A). From the comments to reviewer 1, it seems like this experiment was only performed 1 time. If this is true, the data in Figure 3C should be removed from the manuscript.

3- Line 468-471- Omit these lines. The authors have not provided any data to indicate that Spt5 prevents foreign DNA from being incorporated into the genome. This is not relevant to their study and is superfluous.

4- Lines 428-430: It has not been definitively shown that Spt5 interacts with the non-template DNA to facilitate pausing (and the authors do not provide a reference for this). The authors should rephrase this sentence to state that it likely that Spt5 acts this way and cite this reference: DOI: 10.1016/j.jbc.2023.105106

5- The authors have reduced their introduction and discussion, which is good. These sections are still quite long and would benefit from further trimming.

6- Figure R2 would be good to actually include in the text to show how the structural findings in this manuscript compare to those in other studies.

We were pleased that the reviewers found our work interesting, and we appreciate the opportunity to submit a revised manuscript. The reviewers made excellent suggestions for additional analyses and experiments that we have performed (summarised below). With the addition of new data and other improvements suggested by the reviewers, we believe we have strengthened the main conclusions of the manuscript and hope to have satisfied the reviewers' concerns.

Summary of the analyses:

Additional experiments were performed to further investigate the relationship between Spt5 and Xrn2. Notably, we examined Xrn2 occupancy upon Spt5 loss using ChIP-seq, which revealed a striking reduction in Xrn2 occupancy genome-wide.

To investigate further into Xrn2's interaction with Pol II, we generated an Xrn2 mutant (R492E/R496E) and demonstrated its reduced Pol II binding in vitro. We also created an Xrn2 flexible loop deletion mutant and assessed its occupancy genome-wide, revealing reduced recruitment of Xrn2 during Pol II transcription in the absence of the loop.

We also extended our analyses to provide greater insight into Spt5 function. This included investigating the correlation between upstream readthrough and increased transcription for ncRNAs after Spt5 depletion, analysing the sequence context of the bi-directional promoters activated upon Spt5 depletion (Figures R1-R3), and evaluating the effects of Spt5 depletion in a gene length-dependent manner (Figure S6C).

The manuscript has been revised to accommodate new data and the reviewer's suggestions. We have shortened the Introduction, Discussion, and Results sections to improve focus and clarity. We have also provided additional details on cryo-EM data processing and experimental setup as suggested by the reviewers.

Finally, we have integrated recently published relevant work into our discussion, particularly new cryo-EM structures that complement our findings.

Below we provide a point-by-point response to the reviewers' questions and suggestions. The reviewers' comments are in regular font, and our responses are in italics. To make it easier for the reviewers to locate changes or avoid confusion if figure numbers or panels have been changed in the revised manuscript, our responses provide information on what was changed and explain how previous figures correspond to the figures in the current version.

Reviewer #1 (Remarks to the Author):

The manuscript by Kus et al, reports a new interaction between Spt5, Xrn2 and RNA polymerase II (PolII) as well as a stimulatory role of Spt5 on Xrn2 activity. They then pursue their analyses through the characterization of the role of Spt5 and Xrn2 in transcription regulation. While the initial findings reported are new and of interest, the present manuscript generally lacks in novelty, reporting mostly previously known roles of Spt5 and Xrn2. Additionally, the writing is lengthy and at times confusing, making it

difficult to properly assess whether the authors are trying to develop new models or are barely describing previously established ones. For these reasons, further detailed below, the manuscript does not seem suitable for publication in Nature Communication.

We thank the reviewer for their comments and appreciate the acknowledgment that the data linking Spt5-Pol II and Xrn2 are interesting and new. We have now performed additional experiments that we believe address the key points raised by the reviewer. We have also rewritten relevant parts of the manuscript to improve focus and clarity.

We respectfully disagree with the reviewer regarding analyses of the transcriptome upon Spt5/Xrn2 depletions. Although a previous study by Fred Winston lab performed ChIP-seq and RNA-seq analyses of the transcriptome upon Spt5 depletion, our methodology (including depletion timing and the resolution provided by TT-seq) as well as the analyses of the data extend much further beyond the reported data, providing novel insights. These insights include systematic analyses of the non-coding transcriptome, contributions to transcription termination, 3' end processing, Spt5 involvement in attenuation at the 5' end of the genes, and sequence biases and transcriptional interference and RNA processing defects in Xrn2 catalytic mutant.

Additionally, we have now probed the link between Spt5 and Xrn2 by investigating the importance of Pol II and Spt5 in Xrn2 recruitment during transcription.

Major points:

- While the reported Spt5-Xrn2-PolIII interaction and the stimulatory role of Spt5 on Xrn2 activity are both new, these findings are not further investigated while instead experiments reproducing previously known results are privileged.

Experiments to consider could be to:

- investigate the impact of Spt5 on Xrn2 activity in vivo, or a more direct comparison of the phenotypes resulting from individual depletion of these factors.

As suggested by the reviewer, to address the importance of Spt5 for Xrn2 function, we examined Xrn2 occupancy upon Spt5 loss. Interestingly, acute loss of Spt5 led to a striking reduction in Xrn2 occupancy genome-wide (Figures 5I, J and K), highlighting the importance of Spt5 for the co-transcriptional recruitment of Xrn2. In parallel, we also analysed Pol II occupancy genome-wide by ChIP-seq to assess how Spt5 loss impacts the Pol II profile.

*Although Spt5 and Xrn2 co-occupy Pol II, as evidenced by our purification of native complexes and reconstitution experiments, and there is a close functional link between these factors suggested by our data, the mechanism involved in the activation of Xrn2 by Spt5 is not fully clear and remains a subject for future research. Interestingly, recently published Cryo-EM structure of the *S. cerevisiae* PEC with Rat1/Rai1 (Zeng et al., 2024, Nature;) show that Xrn2 is in proximity to the Kow5 domain of Spt5 near the RNA exit channel. This finding is consistent with our cross-linking data and supports our observations that the Kow5-CTR stimulates Xrn2 enzymatic activity.*

However, while we observe cross-links between Kow5-CTR and Xrn2, the Kow5 domain does not appear to interact directly with Xrn2 in the Cryo-EM structure, and the CTR is not visible. Although further exploration of the mechanism by which Spt5 stimulates Xrn2 activity is beyond the scope of this study, gaining additional insights will enable a more in-depth assessment of the functional contribution of the Spt5-Xrn2 link to transcription.

- mutate the small helical region (481-496) of Xrn2 reported to mediate the interaction with Spt5, either in the R492/496 or I489.

To address the point raised by the Reviewer 1 (also detailed in our response to point 6 raised by Reviewer 3), we generated the Xrn2 R492E/R496E mutant and performed an in vitro pull-down experiment comparing Pol II binding for Xrn2 WT and the Xrn2 R492E/R496E mutant (Figure 2F). As predicted from our structural data, the results revealed that the Xrn2 R492E/R496E mutant shows reduced Pol II binding compared to Xrn2 WT, suggesting that these residues indeed contribute to the interaction between Xrn2 and Pol II.

To examine whether the flexible region is important for mediating the recruitment of Xrn2 during transcription, we replaced the region between amino acids 444 and 555 of Xrn2 with a short linker (GSASGASGG) and evaluated Xrn2 occupancy genome-wide using calibrated ChIP-seq. Interestingly, the loss of the flexible loop led to reduced occupancy of Xrn2 globally, supporting the idea that this region participates in the interaction with Pol II and the recruitment of Xrn2 during Pol II transcription (Figures 2G and H).

We have now incorporated these data into the Results and Discussion sections of the manuscript.

- The results reported in Figure 3D, and the whole Figures 4 and 5 merely reproduce previously reported and well characterized roles of Xrn2 and Spt5. Curiously, the authors seem to be aware of this referencing previous literature.

We respectfully disagree with the reviewer as our analyses and the approach provide novel insights over published work as well as forms the basis for exploring the role of Spt5 in transcription in greater depth, including Spt5-dependent Pol II 5' attenuation (Figures 5C, D and S6E) and its correlation with T-rich sequences (Figure S6D). The use of TT-seq provided better sequencing resolution to uncover defects at the 5' end when Spt5 is missing. As a result, we report on the role of Spt5 in the attenuation of non-coding transcription (Figures R1, R3A-C, S6B, F, G, H and L), possibly via premature termination at the 5' end of genes, and systematically assess the effects of factor depletion on 3' end processing (Figures 5G, H, F, B, 4D, E, S6K, M, S5F and G), splicing (Figures S6J and N), and the degree of transcription interference genome-wide (Figures R1, 4E, 4F, S5H and S6O).

Additionally, the AID system we employed achieves faster depletion of Spt5 compared to previous studies that we reference, improving the likelihood of detecting the direct consequences of the depletion of these essential factors.

Importantly, we also extended the analyses to provide greater insight into the functions of Spt5 and Xrn2 (Figures 2F, G, H and 5I, J, K, which further strengthens and complement our conclusions from the in vitro studies.

- The manuscript is written in a very lengthy manner, in particular the introduction and the discussion being overtly lengthy. Focus would be good. The writing style would also profit from careful proofreading. Examples of such text related issues are listed in a specific section at the end of this document.

We have re-written the Introduction and Discussion parts of the manuscript to make it more focused.

- While the authors conclude appropriately from the individual experiments, their broader interpretations (e.g. in the discussion) are sometimes overconclusive or too speculative.

For instance, in the discussion (l.645-646), the authors state that “We provide evidence that Xrn2 forms a stable complex with Pol II which supports the idea of Xrn2 travelling together with elongation complex”. While they indeed establish the existence of an interaction between Xrn2 and PolII, none of their data suggest Xrn2 would be traveling with PolII.

We have extended our analysis to probe Xrn2 occupancy using ChIP-seq. Now we provide evidence that Xrn2 also is found in gene body of many coding genes, while showing peaks towards the 3'end of the genes (Figures 2G and H). Role of Xrn2 during elongation has been also suggested by other studies (doi: 10.1101/gad.350004.122). Since Xrn2 binding to Pol II is mutually exclusive with spliceosome binding and binding of elongation factors (Spt6 and Kow1/2/3/4 and NGN domains of Spt5) it may form stable complex with Pol II after PAS transition. We have clarified this in the relevant sections (pages 13-14).

Minor Points:

- Regarding transcription units (TUs) displaying increased gene body signal upon Spt5 depletion, the authors mention an effect of signal arising from upstream loci which can indeed be visualized in Figure 4E.

We have performed additional analysis that indicates only fraction (cluster 3, n=355) of TUs displaying increase gene body signal is associated with the possible readthrough from the upstream region, however the majority of the increased transcripts (clusters 1 and 2, 684+87 TUs) are not likely to be associated with the read-through from the upstream gene (Figure R1). In contrast, Xrn2 depletion shows

extremely severe readthrough transcription that continues into lowly expressed TUs (cluster 2, Figure 4E) leading to an increase in TT-seq signal (Figures 4E and F).

Did they additionally check for TU lengths? Since a lack of Spt5 induces defects in TSS proximal pausing, polymerases might be able to run through short TUs while they would fail to elongate fully on longer loci, as reported upon depletion of the Integrator complex (Stein et al., 2022).

Indeed, as pointed out by the reviewer it is interesting to test whether longer TUs are more affected by Spt5 depletion. We have performed this analysis (Figure S6C). As predicted, depletion of Spt5 had more pronounced effect on transcription elongation of the longer protein coding genes.

- “Spt5 is required for transcription termination” (l518). The authors seem to adhere to the previously proposed model that Spt5 is required for Transcription Start Sites (TSSs) proximal quality control and that in its absence, at some TUs, polymerases are not able to properly perform splicing (Figure 5G) or terminate a Transcripts End Sites (TESs) (l.543-545). Can this be directly called a role in transcription termination, as it could then be applied to many quality control mechanisms? This title would be more relevant if a direct dependency on the stimulatory effect of Spt5 on Xrn2 activity was established in this specific context, which is unfortunately not the case.

We do not share the same view on this point raised by the reviewer. First, we have demonstrated using a well-defined, reconstituted in vitro system that indeed Spt5 directly stimulates Xrn2 activity. Second, we showed that Xrn2/Rai1 complex is one of the most highly enriched interactors associated with native Spt5 in vivo. In contrast, there were no splicing components or components of the 3'end processing machinery in Spt5 purifications, likely reflecting the indirect nature of the Spt5 contribution to other RNA processing events. Third, we now demonstrate that Spt5 depletion led to reduced recruitment of Xrn2 during transcription (Figures 5I and K). Together, these data demonstrate that there is a clear functional link between Spt5 and Xrn2.

We also note that Spt5 has not been discussed in the context of promoter-proximal checkpoint monitoring RNA processing even though splicing and 3'end read-through defects have been observed upon Spt5 depletion by previous studies. Published literature discussed the role of the Integrator complex, ARS2, and BRD4 in promoter-proximal quality control of RNA processing.

Finally, the section that the reviewer comments on ('Spt5 is required for transcription termination', page 12) is indeed focused on defective transcription termination observed upon Spt5 depletion, which is reflected in the title. The splicing defect and role of Spt5 in 'licensing' of the elongation complex for RNA processing are discussed in the previous section ('Spt5 is required for restricting non-coding transcription and 'licensing' of Pol II complexes at genic promoters', page 11).

- Regarding the sequencing data library generation, the authors specify using a “NEBNext Ultra II Directional RNA Library Prep Kit for Illumina (NEB)” according to recommendation.

This kit seems to have two possible workflows: either “polyA mRNA” or “rRNA depletion”. Providing additional information on usage would be good. Indeed, the authors provide ncRNA based conclusions suggesting they did not rely on the “polyA mRNA” workflow but on the “rRNA depletion” one (as also seen in Figure S4E (in current version - Figure S5E). However, Figure 4F (in current version-Figure 4G), does show rRNA data with a track scaling of 3 million (while other TUs have a scale of a few hundred). Some leftover rRNA might be present, even after its depletion, but this difference in scale seems confusing. It also poses a risk of rRNA taking up most sequencing reads.

We did not perform rRNA depletion therefore we retain information about rRNA. Indeed, rRNA constitutes considerable fraction of sequencing reads but given small S. pombe genome our sequencing depth can produce high quality results.

- In many instances where the TTseq signal is shown, it is truncated (Figure 4C, 5E, S4G, S5F, S5H, S5L) (corresponds to current Figures 4D, 5E, S5G, S6F, H, L), this should not be the case. Should the authors want to display the signal with its current scale, they also need to show the whole non-truncated signal (as in Figure 4F (current Figure 4G) for instance). Alternatively, they may use log transformed signals to be able to visualize low signal.

We appreciate the comment and incorporated suggestions on the figures (or provided figures with not truncated signal in supplement (Figure S6K to Figure 5E and Figure S5F to Figure 4D).

- Figure S1C: the results text suggests that the processivity of PolIII is increased, corresponding with reduced pausing events. This is not clear to me. Have the results been repeated and quantified?

The experiment presented in Figure S1C was repeated using various templates and protein batches and it is highly reproducible. We now also include quantification of the results presented in Figure S1C as relative increase of the run-off product signal between indicated time-points.

- Figure S3A (current Figure S4A): Why is the input RNA still present after 20 minutes of Xrn2 (et al) incubations compared with the data in Fig 1E? Are these assays not identical?

To examine whether Spt5 has any stimulatory effect on Xrn2 activity we employed concentration of the enzyme/substrate to achieve only partial degradation of RNA, in contrast to experiments presented in Figure 1E), where higher amounts of the enzyme were used to achieve complete degradation of RNA.

- Inconsistencies in data representation; Figure 3A and C are the same assay. Why is the lower part of the gel (i.e. the decay product) shown in A and not in C? These experiments would also benefit from quantification, with replicate data.

RNA degradation can be assessed by monitoring either disappearance of the substrate or accumulation of the product. In the experiment shown in Figure 3C degradation product migrated as dye therefore we omitted to avoid confusion. Nevertheless, disappearance of full-length RNA is consistent with Figure 3A. We have provided quantification of the degradation assay (added to the gel in Figure 3A and corresponding Figure legend). Degradation assays were performed at least in duplicate (except for the experiment in Figure 3C due to complex experimental set up related to assessment of the amount and phosphorylation status of Spt5 (Figure 3B)).

To study degradation kinetics we have developed and employed quantitative FA assays that were performed in triplicates (Figures 3D and S4E).

- Figure 4B (current Figure 4C): adding some statistical measurement would be good.

We have now included statistical analysis (Figure 4C and Table S5).

- The order of results can be a confusing at times. E.g. catalytic activity of Xrn2 (in vitro, Figure 3D (current Figure 4A) should be linked to the in vivo data (Figure 4). The mapping of Spt5 in-between disrupts that flow.

We agree with the reviewer that it is more logical to combine in vitro and in vivo experiments examining role of Xrn2 activity for dislodgement of Pol II from DNA and impact on the transcriptome. We have now moved panel from Figure 3D to Figure 4 (Figure 4A) and adjusted references to Figure 3 and 4 in the text to reflect this.

- Lines 442-443 – 3' reads. Should this be 3'-extended reads? The data is not 3' RNAseq. This sentence was confusing.

Corrected.

- Spt5 rapid depletion TTseq 'highly reproducible'. This is based on only two replicates.

Corrected.

- Line 613 – I'm not sure if it can be said that 'transcription' is increased. TT-seq is a semi-transcriptional technique, but it is not a direct measure of transcription.

Corrected.

Text specific issues:

- The authors tend to overuse the definite article (THE) e.g. 'the core Pol II', 'the nascent RNA',...

We have proofread the text to improve usage of articles.

- Some sentences seem surprising, e.g. the abstract statement: "Although importance of Pol II transcription in pre-mRNA processing is well established". Which RNA species requiring processing is supposed to exist without transcription? The authors are most likely referring to the coupling between transcription and processing, but this choice of phrasing is rather curious.

We have re-phrased this sentence.

- In many instances, casual language is used, e.g. l.192 'Xrn2 chases down still transcribing Pol II', l.175 '... that Pol II requires help from Spt5...', ...

Corrected.

- Among instances of lengthy text, the authors tend to make additional introduction within the manuscript e.g. in Results section 1 (page 7)

Corrected.

Reviewer #2 (Remarks to the Author):

The authors present a study in yeast showing new insights into the role of Xrn2 and Spt5, and how the latter drives the function of the former. The work aligns well with previous studies and contributes new findings that will be of interest to the gene regulation community. The biochemical work is beautiful and clean which is always a pleasure to see, and the overall presentation of the data is of high quality.

We thank the reviewer for positive comments and finding our study interesting.

I have the following questions and comments to clarify some important aspects of the study:

1. The authors show that Xrn2 would compete for Pol II association with the U1 splicing complex. This means that Xrn2 would have to be dislodged during splicing, which for many genes with multiple introns, would occur many times during a single round of elongation. This means that Xrn2 cannot stay bound to Pol II all the time. The authors also suggest that the binding of Xrn2 may be transient. So, does Xrn2 really latch onto Pol II early on, or is it coming on and off at regular intervals or only during termination after CPA cleavage? This needs clarification.

We propose that Xrn2 might be transiently associated with transcriptional machinery during transcriptional cycle based on ChIP-seq (Figure 2G, H), however formation of the stable complex coupled to the remodelling of Pol II interactions with the elongation factors upon full latching to Pol II and Xrn2 activation is likely to occur in the context of

termination (premature or at the end of the genes). We have now integrated this into the relevant section of the Discussion (pages 13,14).

2. Unless I am mistaken, the binding of Xrn2 would also compete with that of Spt6, based on structural work. If this is true, then Xrn2 binding would interfere with Spt6 activity which would be detrimental for normal elongation. It would not make sense then for Xrn2 to be Pol II associated all across the gene bodies (also due to the competition with splicing machinery mentioned above). In any case, it would be helpful to display the binding of Xrn in the context of the complete Pol II elongation complex structures available from the literature, to get a clearer picture of where Xrn2 binds via a vis others elongation proteins.

Indeed, although the helical region of the Xrn2 that is seen in our structure is not overlapping with Spt6 bound we are confident that core of Xrn2 when engaged with RNA would be incompatible with Spt6 binding. This is supported by S. cerevisiae Pol II-Rat1/Rai1 structure that was published (Zheng et al., /doi: 10.1038/s41586-024-07240-3 and Yanagisawa et al., <https://doi.org/10.1101/2024.03.28.587100>), while we were preparing the revision of our manuscript. In agreement with our data, this study also shows that Xrn2 forms stable complex with Pol II binding near the RNA exit channel, which nicely complements our data. Our study further demonstrates that in addition to forming stable complex with Pol II, elongating complex also contributes to full activation of Xrn2 activity to support efficient termination of transcription. Our findings support the model where engagement with Pol II must precede RNA degradation by Xrn2 challenging the current view on the order of the events during transcription termination. We now integrate and discuss recently published work in our manuscript (Page 14).

We include a figure presenting, modelling position of Xrn2 core based on the published structures (Figure R2).

3. In the termination assay in Fig. 3D (current Figure 4A), the correct control to use would be another unrelated exoribonuclease that doesn't bind Pol II. This would be an important control to show that Xrn2 alone does the job but not another protein with the same RNase activity.

We agree with the reviewer that it would be interesting to test whether another exoribonuclease could dislodge Pol II. Similar type of experiment was performed previously showing that eukaryotic Xrn1 which is cytoplasmic 5'-3' exoribonuclease was sufficient to dislodge bacterial polymerase from DNA template (<https://www.embopress.org/doi/full/10.15252/embj.2019102500>) which suggests that this activity alone can terminate RNAP from DNA-RNA scaffold.

4. Lines 164-166: the authors should cite additional papers showing that depletion of Spt5 leads to termination defects (PMID: 29514850, PMID: 28318822).

We thank the reviewer for suggestion and have now incorporated these additional citations (references 117 and 118).

5. The finding that Spt5 can suppress some non-coding transcription in yeast is very interesting. Could the authors speculate on what makes these promoters different from

the others where Spt5 loss leads to decreased gene body elongation. Do they have high levels of Pol pausing at their TSSs? A Pol II ChIP would address this.

We evaluated non-coding transcripts emerging from divergent promoters that are increased upon depletion of Spt5 (Figure R3A). Comparing genes that exhibit up-regulation of non-coding transcription to unaffected genes, indicated that these promoters have higher Pol II occupancy close to TSS (based on Pol II ChIP-seq that we performed during revision). Additionally, region upstream of Spt5 regulated bi-directional promoters show higher CG content (Figures R3B and C). Nevertheless, more mechanistic and more targeted studies are required to fully understand role of Spt5 in suppression of non-coding transcription.

6. Is the mechanism by which Xrn2 terminates transcription known, meaning, how exactly does it displace Pol II once it reaches the RNA exit channel?

*Although the exact mechanism is not fully understood, recent studies have shed some light on Xrn2 function in transcription termination. What is clear from our and other studies is that nucleolytic activity is required for termination. Also, Xrn2 binding is incompatible with the Pol II interaction with the elongation factor Spt6, suggesting that its recruitment to Pol II may slow down or induce pausing of the elongating complex, contributing to efficient dislodgement of Pol II from DNA. Our structure shows that Pol II itself does not undergo conformation change upon Xrn2 binding like what was observed for *S. cerevisiae* Rat1-Pol II complex by Zeng et al. This study additionally reports two different conformations of Xrn2 on Pol II showing rotated Xrn2 when engaged on a shorter RNA leading to hypothesis that the cleavage and translocation of Xrn2 pulls RNA towards itself facilitating release the RNA from Pol II, which will need to be further tested by the future studies.*

We have now incorporated this into the discussion (pages 13-14).

Reviewer #3 (Remarks to the Author):

Here Kuś et al., present a combination of biochemical, structural, and genomics data to study the role of Spt5 in transcription termination in *S. pombe*. First, the authors present a proteomics analysis of Spt5 interactions and show that the termination exonuclease Xrn2/Rai1 interacts with Spt5 regardless of its phosphorylation state. Next, they used a catalytically inactive mutant of Xrn2 to form a stable complex with Pol II, Spt4-5, and Xrn2/Rai1 and subject it to crosslinking-mass spectrometry and cryo-EM analysis. They describe density in their cryo-EM maps corresponding to a central Xrn2 helix that contacts the RPB2 subunit of Pol II and crosslinking data suggests that another Xrn2 region lies in proximity to Spt5 Kow5. They show that Xrn2 exonucleolytic activity is stimulated by Spt4-5 and particularly by Kow5-sCTR. The authors, then made an auxin inducible degron system to degrade Xrn2 and complement either with WT or catalytically inactive Xrn2. They observe run-on transcription after PAS signal, as previously observed. The defect is efficiently restored by WT Xrn2 but not with the catalytically inactive version of the protein. Finally, they

transiently degrade Spt5 and find several defects in Pol II transcription, including enhanced promoter-proximal accumulation and defects in termination, as well as defects in splicing. The authors conclude, based on their experiments, that Xrn2 forms a stable complex with terminating Pol II (called pre-termination complex, PTC) that enhances Xrn2 exonucleolytic activity and degrades RNA in a 5-3' direction causing Pol II dislodgement and recycling.

The authors present a collection of observations that are consistent with their model, however, their study lacks key controls and in some cases the observations are not clear, or the effects are marginally different from their controls. The manuscript could be divided into two different stories, and this would likely make the story more digestible. Specifically, the overlap between the structural and biochemical data and the genomics data is limited. If the authors could better bridge these two sets of experiments through the creation of specific mutants, this would support the fusion of the two sets of results. This manuscript could be appropriate for Nature Communications after major revisions.

We thank reviewer for the feedback. We have now included additional data in the revised version of our manuscript to address the point raised by the reviewer.

Major comments

1-The introduction and discussion sections are quite long. The manuscript would be greatly helped if the authors trimmed these sections to focus on key points that are relevant to this manuscript. The introduction also orders the findings (final two paragraphs in the introduction) in the opposite order that they are presented in the manuscript. It would be clearer if both orders were consistent.

We have incorporated suggestions: shortened both Introduction and Discussion and changed the order of introducing the findings.

2- The Xrn2 degradation assays are displayed in several ways across the manuscript (e.g., gel-based assays, anisotropy), making it difficult to directly compare experiments. It is also difficult to observe stimulation in their current setup because RNA digestion intermediates are not visible in their gel-based assays (either full RNA or the fully degraded RNA). Densitometry analysis of gels would help make differences between experiments clearer. Sometimes the degradation product is not shown (eg figure 3C), and the time points used between experiments are inconsistent. It is also unclear to the reviewer how many times these experiments were repeated. Consistency between the experiments would greatly enhance interpretability of the experiments.

We employed a combination of complementary approaches to probe Spt5's importance for Xrn2 activity. For clarity, we have indicated for each experiment whether it was carried out in the context of a Pol II assembled complex or with RNA substrate

only. We agree that having the same time points would be easier to follow, but we think that direct comparison between these assays is not fully possible (comparing assays with or without Pol II). Nevertheless, for a given type of assay, we kept the same time points.

As also detailed in response to the Reviewer 1, we have added the quantification (percentage of degradation of the full-length RNA) in Figure 3A. RNA degradation can be assessed by monitoring either disappearance of the substrate or accumulation of the product. In the experiment shown in Figure 3C degradation product migrated as dye therefore we omitted to avoid confusion.

Additionally, we performed an assay in the same format as in Figure 3A, showing Spt5's stimulatory effect on Xrn2 in the absence of Pol II, although kinetics are slightly different due to different experimental setup (with different enzyme/substrate ratio) (Figure R4). We performed gel-based experiments at least in duplicate.

To study degradation kinetics we have developed and employed quantitative FA assays that were performed in triplicates (Figures 3D and S4E).

3- The analysis of XL-mass spectrometry data is incomplete. Cutoffs for XL-mass spectrometry are missing in the supplemental figures, specifically mapping crosslinks onto a Pol II structure to see how many are satisfied by the distance of BS3. Xlink Analyzer is a good tool to use to visualize this. A few representative spectra for the described Xrn2-Pol II crosslinks would also be helpful to provide to show the quality of the data.

As suggested by the reviewer, we have now included additional details for the mass spec analyses. Please refer to Figures S2C and S2D.

4- Cryo-EM data presentation is incomplete. Please include data processing tree, angular distribution plot, FSC curves, local resolution plot of maps used to build the structure, a representative micrograph with scale bar, and representative 2D classes with scale bar. It is likely that the authors did not fully classify their data to find Xrn2 containing particles. From the data analysis description, it appears that no specific classification approach was used to specifically classify for Rpb4/7 or Xrn2. Subtracted classification or focused classification on low pass filtered maps can often yield better results. Here the authors only classified based on their crosslinking data locations.

We have now created a flowchart for the cryo-EM processing, including all the requested validations (Figure S3A). The methods only describe what leads to a final map, omitting what has been attempted during the processing. As the reviewer mentioned, all 3D classification strategies have been performed on binned particles, limiting the resolution to around 5-6 Å, as we found that classification works better when the targeted resolution is limited. Before having the crosslinking data, we were hoping to visualize a full-length Xrn2, which unfortunately was not the case. We tried various strategies to classify out particles containing Xrn2 and the different states of Pol II, even if we did not mention them in the methods section. Among the strategies we used, we first decided to classify the particles with the DNA bubble only. Additional

3D classifications without alignment or 3DVA were performed, either without any focused mask or focusing on the DNA/Spt4, Rpb4/7 only, Spt4/5, or Rpb4/7/Spt5. None of these approaches gave obviously good classes. We could obtain good density for Rpb4/7 in addition to some stripy density next to it that could correspond to either Spt5 or Xrn2, but unfortunately, any further classification or local refinement strategy did not allow us to obtain good enough density suitable for rigid body fitting.

5- Figure 2, Panel E, U1 and Xrn2 are not binding Rpb2 same way. When superimposed, they lie perpendicular to each other. This needs to be reworded. It is possible to say that both proteins use overlapping binding surfaces. It looks like Xrn2 is binding the Rpb2 protrusion. If true (or another Pol II domain), this needs to be more specifically described using Pol II structural nomenclature.

We have now reworded the description and updated the location of the helix.

6- The authors perform no experiments to confirm their cryo-EM results. Mutations in the Xrn2 helix that contacts Rpb2 should disrupt the interaction with Rpb2 and should result in reduced Xrn2 activity in the termination and/or RNA degradation assays. If such mutations are identified, it would be useful to add them into their genomic assays. This would also help bridge the genome wide and biochemical findings.

To address this point, we investigated the importance of the helical loop in Xrn2-Pol II interaction using in vitro and in vivo approaches. To achieve this, we purified recombinant catalytically inactive Xrn2 that harbours point mutations in the Pol II interacting helix (R492E/R496E) and employed it in an in vitro pull-down experiment with Rpb9-Flag-Pol II immobilized on beads (Figures 2F and S3E). This experiment revealed reduced Pol II binding for the Xrn2 R492E/R496E mutant, in agreement with the role of the helix in anchoring Xrn2 to Pol II.

To further test the importance of the Xrn2 linker, we replaced the region between 444 and 555aa of Xrn2 with a short linker (Xrn2^{Δ444-555}) and evaluated Xrn2 occupancy genome-wide using calibrated ChIP-seq. We demonstrate that the Xrn2^{Δ444-555} mutant shows reduced recruitment compared to WT Xrn2. Normalization of Xrn2 ChIP-seq for Pol II levels does not change this conclusion (data not shown). These data confirm our structural findings on the nature of the interaction between Xrn2 and Pol II, supporting the importance of the interdomain linker for Xrn2-Pol II interaction and recruitment during Pol II transcription (Figures 2G and H).

We have also analysed Pol II occupancy in Xrn2^{Δ444-555} mutant cells compared to WT cells by equilibrated Pol II ChIP-seq (data not shown). Although there is some reduction in Pol II occupancy, there is no obvious effect on transcription termination observed in the Xrn2^{Δ444-555} mutant. This could be explained by residual Xrn2 still being able to carry out its function. We also note that the Xrn2^{Δ444-555} mutant shows higher protein levels compared to WT, possibly compensating for a partial loss of function (Figure S3F). Interestingly, disrupting physical coupling of the cytoplasmic 5' to 3' exonuclease counterpart of Xrn2, Xrn1, to translating ribosomes does not appear to

affect Xrn1-mediated mRNA degradation or impact cellular growth (Tesina et al., 2019 NSMB, doi: 10.1038/s41594-019-0202-5), suggesting that under standard growth conditions, cells can cope with the loss of coupling due to other compensatory mechanisms.

We have incorporated new data in the Results and Discussion sections of the manuscript.

7- The authors claim that CDK9 does not affect Xrn2 activity in their assays. Have they tested whether CDK9 is indeed phosphorylating Xrn2? Mass spectrometry analysis would address this issue. Additionally, the authors could make a phosphomimetic mutant based on previous data from the Fisher lab (10.1101/gad.269589.115) as a control.

We apologise that our explanation was not clear, and we rephrased relevant sections. We have investigated how Spt5 phosphorylation affects Xrn2 stimulation. Cdk9 addition alone was used to control for effects beyond Spt5. We have monitored Spt5 phosphorylation using phos-tag gel (Figure 3B) demonstrating that Cdk9 is active. It is not clear whether fission yeast Xrn2 is phosphorylated by CDK9. However, since Xrn2 phosphorylation was proposed to stimulate activity of Xrn2, and we do not observe additional stimulation of Xrn2 when we performed control experiment with Cdk9 (during complex assembly Cdk9 should be washed away), we believe that this point is not relevant and beyond the scope of current manuscript.

Minor comments

1- The authors refer to Spt4-5 as Spt5/4. This is a bit awkward and is non-standard in the field.

We have now replaced Spt5/4 for as Spt4/5 as suggested by the reviewer.

2- Figure 1- some interactors are missing in the IP experiment (e.g., Rtf1). The authors mentioned that is due to a transient interaction, however there is no indication of what the authors consider transient, as some factors are known to be stably bound to Pol II and are still missing in their experiment. The reviewer suggests omitting any reference to transience.

We omitted this reference.

3- The authors claim that the degradation we see is a 5'-3', as expected for the previously described Xrn1 activity, but there is no control for that assumption. A protected 5' RNA would be a useful negative control for this point.

We include a gel showcasing activity of Xrn2 towards 3'-FAM-labelled but not 5'-FAM-labelled RNA (Figure R5).

4- Figure 2, pulldown/IP experiments. It would be good to see the inputs at least in the supplemental figure. Also, some IP combinations are missing, for example Rai1 + Spt4/5. This will help to rule out any unspecific binding to the beads. Also, more labelling of the gels would be helpful (like labelling the position of Rpb1, Rpb3, Rpb5, Spt4, etc.)

As suggested by the reviewer we have labelled bands on the gels (i.e. Figure 2B). To control for unspecific binding to the beads, beads without immobilised Pol II were challenged with Spt5/Rai1/Xrn2 (Figure 2, lane 1). This experiment demonstrates no unspecific binding to the beads.

5- The discussion does not compare their findings in light of alternative models. For example, how are these findings consistent or inconsistent with previous studies supporting the torpedo model?

Following the Reviewer suggestion, we discuss our findings in light of the existing 'torpedo' model in several relevant sections of the manuscript (Abstract, Introduction, Discussion and Figure legend for Figure 6.)

Figure R1**Figure R2****Figure R3****Figure R4****Figure R5**
Figure R1. Upregulated ncRNAs after Spt5 loss have multiple origins. Signals for upregulated ncRNA were classified according to the signal before the annotated TSS (k-means=3). A fraction of ncRNA might have originated from read-through of the upstream gene.

Figure R2. Spt6 binding is incompatible with Xrn2 catalytic core binding to Pol II. Our structural model (PDB: 8QSZ) was overlaid with the model containing Spt6/Spt5/Spt4 (PDB: 7XN7). The *S. pombe* Xrn2 position (PDB: 3FQD) was modeled based on the *S. cerevisiae* Xrn2/Pol II (PDB: 8JCH). RNA paths for different models are highlighted with arrows.

Figure R3. Divergent anti-sense emerges from promoters with certain characteristics. (A) Anti-sense originating from divergent promoters after Spt5 depletion was selected and plotted as a heatmap of relative TT-seq signal (n=244). (B) Genes that generate divergent anti-sense transcription have, on average, higher Pol II occupancy. The metaprofile for genes with upregulated divergent anti-sense is compared to control genes. Intervals around the mean signal correspond to the standard error of the mean and to confidence intervals (using bootstrapping) for anti-sense and control genes, respectively. (C) Comparison of the CG content in the 100 bp window before the TSS for genes with divergent antisense (n=244) or control genes (n=2866). Statistical significance was evaluated with two-sample Kolmogorov–Smirnov test.

Figure R4. Spt5 stimulates Xrn2 exoribonucleolytic activity in the absence of Pol II. Activity towards 3'FAM-labeled RNA was assessed at the indicated time points, and products were resolved on a UREA-PAGE gel (as in Figure 3A).

Figure R5. Xrn2 does not degrade RNA with 5'-FAM-RNA. Xrn2 activity towards 3'FAM-labelled and 5' FAM-labelled RNA was assessed after 6 min, and products were resolved on a UREA-PAGE gel.

We thank reviewers for their input and help in improving our manuscript. We are pleased that reviewers were satisfied with our revisions. We tried to address remaining issues and provide point-by-point responses to their comments. Our response is in blue, italicised font.

Reviewer #1 (Remarks to the Author):

We thank the authors for their rebuttal and revisions to their manuscript. For the most part, they have addressed all our concerns in regard to data with newly performed experiments supporting their conclusions appropriately. However, one major point remains that the writing is very difficult to follow in places – especially in the Introduction and Discussion sections which are incredibly lengthy. The introduction should be much more concise and does not need to introduce things that are not included in the rest of the manuscript. For example, RPB1-CTD phosphorylation has 13 lines in the introduction but is not mentioned anywhere else in the manuscript. Additionally, lines 111-142 describe the findings of the results but includes far too much detail for an introduction summary. Overall, this still needs to be made much more concise. Further, we made the point of the overuse of the definite article (THE) throughout the paper e.g. “The RNA processing factors”. This still needs attention because it has been removed in places where it should remain and vice versa. Below are a few examples, but it should be carefully addressed throughout the manuscript.

Examples where THE should be removed;

Line 43-44; “the bi-directional promoters”

Line 68; “The Integrator”

Line 70; “The RNA processing factors”

Line 96: “The promoters”

Examples where THE should be included;

Line 161; “In the purification of the Spt5 T1E mutant”

Line 166; “...to test wheter the Xrn2/Rai1 heterodimer”

Finally the use of “the RNA enzymes” in the abstract is a bit cryptic and should be reworded. Does this simply mean XRN2?

We thank the reviewer for suggestions, and we have removed most of the section dedicated to CTD phosphorylation. We trimmed the introduction summary. We made some changes to discussion. We tried to address the usage of definitive articles. We have included in the abstract “including Xrn2” in the sentence concerning “the RNA enzymes”.

Reviewer #2 (Remarks to the Author):

The authors have addressed my questions reasonably. I have no major concerns.

We thank the reviewer for positive feedback.

Reviewer #3 (Remarks to the Author):

I appreciate the efforts the authors have taken to address my concerns. There are some outstanding issues that should be addressed prior to publication.

1- Figure 2D, E- Labelling on RNAPII is sparse. Labels are needed for upstream and downstream DNA, Spt5 (not just KOW5), the active site Mg²⁺ needs to be added to orient readers, and different colors/shadings would help in Panel E to distinguish between the indicated RNAPII subunits that are

highlighted. Currently, labels are provided for 4 subunits, but all subunits are colored white, and it is hard to know which ones are which.

We thank the reviewer for the suggestions which are now implemented in the indicated figures.

2- Figure 3C- show in the same manner as Figure 3A (currently, 3C is missing the nucleotide level band (bottom panel in 3A). From the comments to reviewer 1, it seems like this experiment was only performed 1 time. If this is true, the data in Figure 3C should be removed from the manuscript.

We have removed this panel and sections which referred to this experiment.

3- Line 468-471- Omit these lines. The authors have not provided any data to indicate that Spt5 prevents foreign DNA from being incorporated into the genome. This is not relevant to their study and is superfluous.

We have removed these lines.

4- Lines 428-430: It has not been definitively shown that Spt5 interacts with the non-template DNA to facilitate pausing (and the authors do not provide a reference for this). The authors should rephrase this sentence to state that it likely that Spt5 acts this way and cite this reference: DOI: 10.1016/j.jbc.2023.105106

We rephrased this sentence and included supporting citation.

5- The authors have reduced their introduction and discussion, which is good. These sections are still quite long and would benefit from further trimming.

We tried further to trim and shorten mentioned sections.

6- Figure R2 would be good to actually include in the text to show how the structural findings in this manuscript compare to those in other studies.

We have included Figure R2 as Supplementary Fig. 3h.